# ROBUST FUNCTION-CALLING FOR ON-DEVICE LANGUAGE MODELS VIA FUNCTION MASKING

**Qiqiang Lin**[1*]     **Muning Wen**[2*]     **Qiuying Peng**[1*†]     **Guanyu Nie**[3‡]     **Junwei Liao**[2]

**Jun Wang**[1]   **Xiaoyun Mo**[1]   **Jiamu Zhou**[1]   **Cheng Cheng**[1]   **Yin Zhao**[1]   **Jun Wang**[1†]

**Weinan Zhang**[2†]

[1]OPPO Research Institute   [2]Shanghai Jiao Tong University   [3]Iowa State University
{pengqiuying,wangjun7}@oppo.com
wnzhang@sjtu.edu.cn

## ABSTRACT

Large language models have demonstrated impressive value in performing as autonomous agents when equipped with external tools and API calls. Nonetheless, effectively harnessing their potential for executing complex tasks crucially relies on enhancements in their function-calling capabilities. This paper identifies a critical gap in existing function-calling models, where performance varies significantly across benchmarks, often due to being misled by specific naming conventions. To address such an issue, we introduce Hammer, a novel family of foundation models specifically engineered for on-device function calling. Hammer employs an augmented dataset that enhances models' sensitivity to irrelevant functions and incorporates function masking techniques to minimize misleading. Our empirical evaluations reveal that Hammer not only outperforms larger models but also demonstrates robust generalization across diverse benchmarks, achieving state-of-the-art results. Our open-source contributions include a specialized dataset for irrelevance detection, a tuning framework for enhanced generalization, and the Hammer models, establishing a new standard for function-calling performance.

## 1 INTRODUCTION

Large language models (LLMs) have demonstrated remarkable proficiency in addressing a wide range of natural language processing tasks (Chowdhary & Chowdhary, 2020), as well as in handling long-context reasoning and complex planning (Wen et al., 2024). The use of LLMs as autonomous agents to assist humans in completing intricate tasks is increasingly in demand and is now more feasible from a technical standpoint than ever before (Gunter et al., 2024). To fully capitalize on the potential of LLMs as autonomous agents, it is crucial for these models to accurately identify and utilize external tools or application programming interfaces (APIs), thereby enabling them to effectively execute complex tasks (Abdelaziz et al., 2024; Patil et al., 2023). Central to this capability is the model's ability to select appropriate functions from a given set of options, provide accurate input arguments, and ultimately fulfill the user's intent. Furthermore, in scenarios where no suitable function exists within the available options, the model must have the ability to decline the task, rather than making incorrect attempts (Patil et al., 2023). Recent advancements have introduced a variety of relevant datasets and benchmarks (Li et al., 2023; Wu et al., 2024), along with the release of powerful models specifically designed for function-calling tasks (Zhang et al., 2024; Patil et al., 2023; Abdelaziz et al., 2024). Some models even simulate real-world scenarios, such as

---

[*]Equal Contribution.
[†]Corresponding Author.
[‡]Work conducted during an internship at OPPO Research Institute.

Table 1: Inconsistent performance of existing function-calling models across different benchmarks. For example, although xLAM-7B-fc achieved the best performance on most of the benchmarks, its performance significantly declined on the other two, resulting in the lowest average score overall.

| Models | BFCL | API-Bank | SealTool | Tool-Alpaca | Nexus Raven | Avg. |
|---|---|---|---|---|---|---|
| Gorilla-OpenFunctions-v2-7B (FC) | 79.1 | 62.5 | 91.1 | 51.3 | 68.4 | 70.48 |
| Granite-20B-FunctionCalling (FC) | 76.63 | 68.5 | 92.7 | 58.0 | 75.1 | 74.186 |
| xLAM-7B-fc (FC) | 79.41 | 72.45 | 76.9 | 59.0 | 57.5 | 69.052 |

ticketing systems, to mimic more realistic use cases (Yao et al., 2024; Chen et al., 2024a). Despite these significant strides in the development of function-calling models, our investigation reveals a critical gap: *many existing models demonstrate considerable performance variations across different benchmarks*. As illustrated in Table 1, this inconsistency underscores the need for further research into the robustness and generalization of function-calling models across diverse and practical task environments.

Achieving such stability across diverse benchmarks is crucial, as it indicates the model's capability to generalize effectively to real-world applications (Yao et al., 2022; Zhang et al., 2025). Driven by this objective, we conduct a thorough analysis of the instability observed in existing models when executing function-calling tasks. Our findings highlight that one of the primary factors influencing generalization performance across benchmarks is the misleading nature of specific naming conventions for functions and parameters. Consequently, existing models tend to perform well on benchmarks that closely align with the naming conventions in the training data but suffer notable performance declines when encountering benchmarks with differing naming styles. This problem is examined in detail in Section 3, which motivates us to propose the function-masking technique.

In this paper, we present the **Hammer**, a family of lightweight models specifically fine-tuned for on-device function-calling tasks. This work is underpinned by a carefully designed irrelevance-augmented dataset and the use of function masking techniques, both aimed at enhancing the generalization capabilities of the models. To improve the models' ability to determine whether the user's intent aligns with the available function calls, we augment the xLAM-function-calling-60k dataset (Liu et al., 2024b) with an additional 7,500 instances specifically tailored for irrelevance detection. Furthermore, we introduce a function masking technique, which shifts the models' focus from function and parameter names to their descriptions, effectively reducing potential misinterpretations.

Following these advancements, Hammer demonstrates robust function-calling performance and strong generalization across a variety of benchmarks. Despite containing only 7 billion parameters, Hammer outperforms many larger open-source models and competes with top-tier closed-source models, such as GPT-4 (Achiam et al., 2023) and GPT-4o (Islam & Moushi, 2024), on the Berkeley Function Calling Leaderboard (BFCL) v2 (Yan et al., 2024). We benchmark Hammer and other models, including Salesforce's xLAM series (Zhang et al., 2024) and IBM's Granite-20B-FunctionCalling (Abdelaziz et al., 2024), across a range of representative datasets, such as API-Bank (Li et al., 2023), Tool-Alpaca (Tang et al., 2023), Seal-Tools (Wu et al., 2024), and Nexus Raven API Evaluation (Srinivasan et al., 2023). The results consistently highlight Hammer's exceptional generalization capabilities. The key contributions of our work could be summarized as follows:

- **Tuning Framework:** A straightforward yet effective framework evolving function masking to tune function-calling models toward robust generalization capabilities.

- **Augmented Dataset:** A specialized dataset with 7,500 instances designed to enhance language models' awareness of irrelevance between candidate functions and user intent.

- **Consistent SOTA Models:** Hammer, a family of well-trained function-calling models that demonstrate state-of-the-art performance across multiple benchmarks.[1]

## 2 RELATED WORKS

**LLMs as Agents for Function-Calling.** Recent research has shown significant interest in leveraging LLMs as autonomous agents to perform complex tasks through function calling and tool usage (Erdogan et al., 2024a; Chen et al., 2024b). IBM Granite-20B-FunctionCalling model (Abdelaziz et al., 2024) proposes a multi-task learning framework trained on seven core function-calling tasks, demonstrating superior performance over other open models on the BFCL v2 benchmark. APIGen

Table 2: Performance of different models on Berkeley Function-Calling Leaderboard (as of date 09/20/2024). The rank is based on the overall accuracy, which is a weighted average of different evaluation categories. "FC" stands for function-calling mode in contrast to using a customized "Prompt" to extract the function calls. For the complete list, refer to Table 10 in Appendix E.

| Rank | Model | Overall Acc | AST Summary | Exec. Summary | Irrelevance | Relevance |
|------|-------|-------------|-------------|---------------|-------------|-----------|
| 1 | GPT-4-0125-Preview (Prompt) | 85.79 | 85.50 | 89.25 | 61.35 | 97.56 |
| 2 | GPT-4-1106-Preview (Prompt) | 85.00 | 86.31 | 87.38 | 64.98 | 90.24 |
| 3 | GPT-4-0613 (Prompt) | 84.74 | 84.66 | 87.57 | 75.57 | 82.93 |
| | Hammer-7B (FC) | 83.92 | 78.70 | 89.72 | 72.87 | 92.68 |
| 4 | GPT-4-turbo-2024-04-09 (Prompt) | 83.89 | 85.41 | 88.13 | 61.82 | 82.93 |
| 5 | GPT-4o-mini-2024-07-18 (Prompt) | 83.35 | 80.52 | 87.95 | 79.20 | 80.49 |
| 7 | Functionary-Medium-v3.1-70B (FC) | 82.55 | 81.06 | 89.32 | 73.23 | 70.73 |
| 13 | Functionary-Small-v3.1-8B (FC) | 80.21 | 78.64 | 83.45 | 68.36 | 85.37 |
| 16 | xLAM-7B-fc (FC) | 79.41 | 72.77 | 85.68 | 79.76 | 80.49 |
| 19 | Gorilla-OpenFunctions-v2-7B (FC) | 79.10 | 73.18 | 84.97 | 73.13 | 85.37 |
| 21 | Functionary-Small-v3.2-8B (FC) | 78.96 | 76.16 | 83.04 | 72.32 | 80.49 |
| 25 | FireFunction-v2-70B (FC) | 77.45 | 74.20 | 84.23 | 52.94 | 87.80 |
| 26 | Granite-20B-FunctionCalling (FC) | 76.63 | 66.73 | 82.97 | 72.43 | 95.12 |
| | Hammer-4B (FC) | 76.05 | 69.59 | 80.82 | 68.66 | 90.24 |
| 31 | xLAM-1.3B-fc (FC) | 74.90 | 67.37 | 80.80 | 61.21 | 95.12 |
| 32 | Hermes-2-Pro-Llama-3-70B (FC) | 74.78 | 72.09 | 81.29 | 53.80 | 80.49 |
| | Hammer-1.5B (FC) | 73.04 | 65.53 | 75.86 | 72.18 | 92.68 |
| 40 | Command-R-Plus (FC) | 72.04 | 66.32 | 77.41 | 52.75 | 92.68 |
| 45 | Hermes-2-Pro-Llama-3-8B (FC) | 66.18 | 64.18 | 74.05 | 55.16 | 53.66 |
| 46 | Hermes-2-Pro-Mistral-7B (FC) | 65.44 | 60.82 | 74.25 | 38.55 | 75.61 |
| 47 | Hermes-2-Theta-Llama-3-8B (FC) | 64.83 | 61.08 | 72.54 | 62.66 | 51.22 |
| 57 | FireFunction-v1-46B (FC) | 48.11 | 38.16 | 41.20 | 68.55 | 95.12 |

Table 3: Performance comparison of different models on several academic benchmarks. The rank is based on the average F1 score on "Func. + Args", which indicates both function selection and parameter filling are accurate. For intermediate details see Table 11 in Appendix E.

| | Academic Benchmarks (F1 Func. + Args) | | | | | |
|-------|------------|------------|-------------|-------------------------|-------------|---------------|
| Model | API-Bank L-1 | API-Bank L-2 | Tool-Alpaca | Seal-Tools (Single-Tool) | Nexus Raven | F1 Average |
| GPT-4-0613 (Prompt) | 84.78 | 56.98 | 66.67 | 93.95 | 91.60 | 78.79 |
| GPT-4o-mini (Prompt) | 89.28 | 67.52 | 54.69 | 86.00 | 84.59 | 76.42 |
| Hammer-7B (FC) | 85.79 | 66.40 | 59.86 | 91.66 | 77.35 | 76.21 |
| Granite-20B-FunctionCalling (FC) | 77.82 | 59.15 | 58.00 | 92.70 | 75.14 | 72.56 |
| Hammer-4B (FC) | 81.46 | 61.01 | 56.96 | 92.45 | 64.89 | 71.35 |
| xLAM-7B-fc (FC) | 80.69 | 64.24 | 58.96 | 76.87 | 57.50 | 67.65 |
| Gorilla-OpenFunctions-v2-7B (FC) | 70.34 | 54.69 | 51.26 | 91.11 | 68.41 | 67.16 |
| xLAM-1.3B-fc (FC) | 83.70 | 64.32 | 50.58 | 80.43 | 54.80 | 66.77 |
| Hammer-1.5B (FC) | 72.30 | 59.71 | 53.48 | 88.65 | 56.88 | 66.20 |
| Qwen2-7B-Instruct (Prompt) | 60.62 | 49.50 | 48.11 | 77.51 | 63.47 | 59.84 |
| Qwen2-1.5B-Instruct (Prompt) | 63.55 | 33.62 | 45.25 | 75.49 | 45.46 | 52.67 |
| Qwen1.5-4B-Chat (Prompt) | 59.78 | 38.48 | 16.98 | 62.32 | 33.70 | 42.25 |

adopts an automated pipeline for generating high-quality, diverse function-calling datasets, building a 7B-parameter model to surpass GPT-4's performance (Liu et al., 2024b). Similarly, ToolACE (Liu et al., 2024a) generates diverse tool-learning datasets, allowing its 8B-parameter ToolACE-8B model to achieve state-of-the-art results on the BFCL v2, rivaling the latest GPT-4 models. Further studies explore various dimensions of function calling, such as improving efficiency through parallel function calls (Zhang et al., 2016), identifying vulnerabilities in function calling processes (Srinivasan et al., 2023), and developing benchmarks to evaluate LLMs' ability to handle diverse function calls (Kim et al., 2024). Collectively, this body of work emphasizes the role of function calling in enabling LLMs to act autonomously and integrate external tools and resources effectively.

**Datasets and Benchmarks for Function-Calling Evaluation.** Substantial advancements have been made in developing datasets and benchmarks to assess the function-calling capabilities of LLMs. API-BLEND (Basu et al., 2024) introduces a large corpus for training and systematically testing tool-augmented LLMs. It includes real-world scenarios involving API-related tasks such as API/tool detection, slot filling, and sequencing of detected APIs. API-Bank (Li et al., 2023) provides a comprehensive dataset featuring 2,138 distinct APIs and 1,888 dialogues with 4,149 API calls. This

---

[1]The source code is available at `https://github.com/MadeAgents/Hammer`, while the augmented dataset and models can be accessed at `https://huggingface.co/MadeAgents`. The latest release, the Hammer 2.1 models, significantly improves multi-turn and multi-step function calling capabilities.

Figure 1: Demonstration of a simple function-calling process.

dataset is designed to evaluate LLMs' tool-utilization capabilities, including planning, retrieval, and API-calling proficiency. APIGen (Liu et al., 2024b) employs an automated and rigorous data generation process to create a diverse dataset that includes various query styles, such as parallel function calling, and undergoes a multi-stage verification process to ensure data accuracy and relevance. Seal-Tools (Wu et al., 2024) introduces a large-scale, self-instruct API-like tool-learning dataset that incorporates practical application scenarios and nested tool calls.

**Tuning Techniques for Function-Calling Models.** IBM's Granite-20B-FunctionCalling model is trained using a multi-task learning approach, which enables language models to develop function-calling capabilities by mastering a range of granular tasks (Abdelaziz et al., 2024). TinyAgent focuses on equipping small language models (SLMs) with complex reasoning and function-calling abilities, allowing for secure and private deployment at the edge. It employs LoRA fine-tuning, incorporates negative samples, and uses in-context examples selected via retrieval-augmented generation (RAG) (Gao et al., 2023) to enhance function selection and orchestration accuracy through directed acyclic graph (DAG) comparison (Erdogan et al., 2024b). The xLAM series utilizes a supervised fine-tuning (SFT) approach with direct preference optimization (DPO) (Rafailov et al., 2024) alignment, integrating data parallelism, LoRA, and a cosine learning rate scheduler to optimize performance across various categories of function-calling agents (Zhang et al., 2024).

## 3 PROBLEM STATEMENT AND ANALYSIS

This section aims to introduce and analyze the common challenges that function-calling models encounter in practical applications. Through this analysis, we seek to identify methods to enhance models' stability and generalization capabilities in real-world scenarios.

Before delving into the specific issues, we present a typical function-calling process, illustrated in Figure 1. In this process, each candidate encompasses several components, including the function name, parameter names, default values, and descriptions. The objective of the model is to output complete and accurate function-calling code that can accomplish users' intent or, alternatively, output an empty list to indicate that none of the given candidates can satisfy the user's requirements (Yan et al., 2024). Achieving this goal hinges on the model's ability to accurately align the user's intent with the functionality of the candidate functions, i.e., selecting the appropriate function, and its capacity to comprehend the usage of each parameter, i.e., populating the function with the correct arguments. However, certain recurring issues have been observed in practice.

### 3.1 MISLEADINGNESS BY FUNCTION NAME AND PARAMETER NAME

As illustrated in Figure 1, the definition of a function typically comprises the function name, parameter names, and descriptions. The format of function and parameter names is often quite compact, e.g., cal_sum or max_value, and influenced by the designer's personal style and preferences. When a model attempts to infer the function's purpose solely from the function name, this compactness can lead to ambiguities, misguiding the model's selection, particularly in the presence of complex functionalities (Gunter, 1992). For instance, a function named `parse_data` might be intended for parsing JSON data, but the same name could refer to parsing CSV files in a different context, leading to potential misinterpretations. Similarly, when deducing the usage of parameters based on their names, models may be misled by the historical usage of similarly named parameters in the training dataset. More specifically, these misleading scenarios can be categorized into several cases.

**Misled by Function Names.** When a user intent aligns closely with a function name present in the training labels, the model may incorrectly prioritize that function from the candidate list during

testing, even if its functionality diverges significantly from the intended operation. For example, if a function named `fetch_data` is included in the training pairs for retrieving user data from a database, but in the testing set, a function with the same name retrieves data from an external API, the model may erroneously select it based solely on the name.

**Misled by Parameter Names.** When the functionality and descriptions of parameters change within the testing environment, the model frequently clings to its original patterns of parameter usage, resulting in incorrect function calls. For instance, if a function's parameter `timeout` is expected to be an integer representing seconds in one context, but in another context, it is defined as a string in the format "10s", the model's reliance on the original integer format may lead to erroneous calls.

**Disturbed by Naming Preferences.** The model's robustness can diminish when the naming conventions of functions or parameters in the testing environment diverge from those in the training dataset. Variations, such as discrepancies between `CamelCase` and `snake_case` may adversely lower the model's confidence, as an on-device lightweight model may struggle to generalize across different naming styles.

### 3.2 THE IMPACT OF EXCESSIVE FOCUS ON THE NAMING

To investigate the extent to which existing models rely on function and parameter names and corresponding impact, we conducted a case study using the xLAM-1B-fc model on the Seal-Tools benchmark. Specifically, we masked the function and parameter names in the test set, i.e., replaced them with random strings, and observed how the model's performance changed. As shown in Figure 2, after masking the function and parameter names, even though the descriptions contained all necessary information about the function's purpose and usage, the performance of xLAM-1B-fc dropped significantly. This result confirms the model's overreliance on function and parameter names, highlighting the potential risks this behavior may pose in real-world applications.

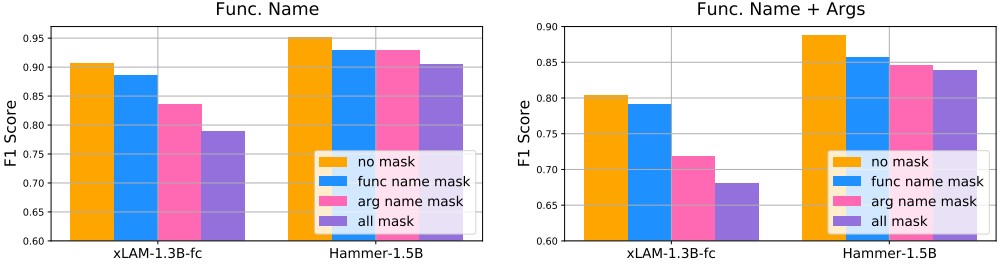

Figure 2: Case studies examining the performance degradation when function names and parameter names are obfuscated during test time. For a detailed analysis of the error cases, see Appendix B.

In contrast, Figure 2 also presents the performance of our Hammer model under the same setting. Hammer exhibited a much smaller performance drop, demonstrating its robustness when faced with arbitrary function and parameter naming patterns. This resilience suggests that Hammer relies more heavily on the function descriptions rather than compact, potentially ambiguous names. In Section 4, we will provide a detailed explanation of Hammer's training methodology.

## 4 METHODOLOGY

In this section, we describe our detailed approach and augmented dataset to fine-tune the Hammers, a series of robust language models designed for function-calling.

### 4.1 FUNCTION MASKING

In light of the analysis in Section 3, a direct approach to mitigate these issues involves minimizing the interference from function names and parameter names, while enforcing the model to comprehend the functionality and usage of candidates based on their descriptions. In contrast, the descriptions provide a more flexible natural language explanation, often encapsulating the information that function and parameter names aim to convey. While descriptions can also reflect the designer's personal style to some extent, they tend to be more accurate and detailed, thus reducing the likelihood of ambiguity or misguidance. Consequently, when training function-calling models, we face

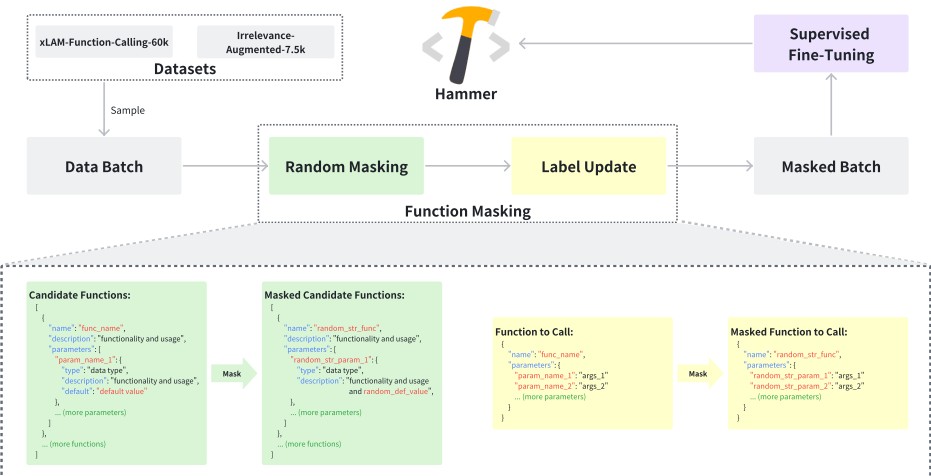

Figure 3: Step-by-step building workflow of Hammer series with function masking. Considering the complexities of real-world scenarios, the masking operation does not apply to all samples. Instead, Hammer enjoys a masking ratio of $0.33$ before each training epoch, which yields the best overall performance across all benchmarks.

the challenge of not knowing the preferences or naming styles of function designers in real-world applications. Thus, it is reasonable to expect that the trained model should understand the function's purpose and usage through its description rather than attempting to infer functionality based on potentially ambiguous, compact components such as function and parameter names.

To this end, we propose a tuning framework for function-calling models based on a masking mechanism, with a full pipeline shown in Figure 3. As for the randomization process, we first randomly determine the length $L \in (5, 15)$ of the random string, and then randomly select $L$ characters from a set comprising 52 uppercase and lowercase letters and 10 digits to replace the corresponding names. This framework aims to guide the model's attention toward the description, and thus enhance the model's generalization capabilities in practice. Specifically, in our proposed framework:

- **Function names** in candidates are masked by replacing them with randomly generated strings during training. This technique minimizes the model's reliance on memorizing function names, prompting it to understand the function's purpose solely through its description. By doing so, the model becomes more adaptable across various coding practices, as it is less influenced by common naming conventions.

- **Parameter names** in candidates are substituted with random strings as well, ensuring the model focuses on the parameter descriptions rather than the specific names, which often vary between implementations.

- **Default parameter values** in candidates are randomized and appended to the parameter descriptions. This also guides models to pay more attention to the parameter descriptions.

- **Labels** in batch are updated according to the masked candidate list, i.e. replacing the function name and parameter name with corresponding masked strings in candidates.

By focusing on the description, the model can more accurately grasp the function's intent and expected behavior, ensuring robust performance across diverse naming conventions and avoiding pitfalls introduced by overfitting to specific naming patterns in the training data. It also promotes better generalization, as the description typically offers a more comprehensive view of the function's role, beyond what can be conveyed by concise names.

## 4.2 IRRELEVANCE-AUGMENTED DATASET

During the fine-tuning process using the xlam-function-calling-60k dataset, we identified a concerning inverse relationship between the model's ability to accurately execute function calls and its capacity for irrelevance detection—specifically, the ability to assess whether there exists no function call in the candidate set aligns with the user's intent. This observation is analyzed further in Section 5.6 and Appendix G. It indicates that, while fine-tuning lightweight language models on

datasets specialized in function selection can improve their accuracy in choosing appropriate functions from a predefined set, it may unintentionally impair their ability to detect irrelevance. As a result, models might generate inappropriate function calls, even in the absence of valid options.

To address this issue, we propose an irrelevance-augmented dataset. This augmentation, applied to the original xlam-function-calling-60k dataset, incorporates 7,500 examples sampled from the original training set. In constructing this dataset, we removed the correct function from the candidate list for each sampled example and replaced the labels with empty lists, indicating that all candidates are irrelevant. This approach ensures that any potential data contamination is limited to what already exists within the xlam-function-calling-60k dataset, as the process does not introduce additional data beyond the original dataset. More details about the data contamination analysis refer to Appendix C.

## 5 EVALUATION

### 5.1 EXPERIMENTAL SETUP

In this section, we show the superiority of our Hammers in performance and robustness across various benchmarks as well as in-depth analysis to verify the effectiveness of our augmented dataset and approach. Besides, potential limitations of function-masking could be found in Appendix D.

**Benchmarks.** To assess the generalizability of Hammers, we conducted evaluations using a variety of function-calling benchmarks, all of which represent out-of-domain challenges for our model. The **Berkeley Function-Calling Leaderboard (BFCL)** (Yan et al., 2024) provides a comprehensive dataset comprising over 1,700 instances. It covers tasks such as Simple Function, Multiple Function, Parallel Function, and Parallel Multiple Function for Python, as well as function relevance detection, REST API, JavaScript, and Java for non-Python environments. **API-Bank** (Li et al., 2023), consisting of 314 tool-use dialogues and 753 API calls, evaluates models' ability to correctly invoke a known API (L-1) based on a query, and to retrieve and call APIs from a candidate list (L-2). Similarly, **Nexus Raven API Evaluation** (Srinivasan et al., 2023) offers 318 test examples across 65 distinct APIs, contributing further to the evaluation of function-calling capabilities. **Tool-Alpaca** (Tang et al., 2023) employs a synthetic data generation method, featuring 271 tool-use instances in 50 categories. For evaluation, we utilized 100 simulated test examples from this dataset, similar to Nexus Raven. Lastly, **Seal-Tools** (Wu et al., 2024) represents one of the most extensive and recent benchmarks, with 4,076 automatically generated APIs across various life domains. As one of the newest benchmarks, Seal-Tools presents a relatively lower risk of data leakage. As for the **Tool-Bench** (Qin et al., 2023), we chose not to utilize it because of potential data contamination risks. Specifically, the xlam-function-calling-60k dataset which serves as our training set, was generated using ToolBench (Liu et al., 2024b). This poses a significant risk of data contamination if ToolBench is used as a benchmark, potentially leading to unfair advantages compared with certain baselines.

**Evaluation Metrics.** BFCL assesses function-calling models through two primary evaluation methods: Abstract Syntax Tree (AST) Evaluation and Executable Function Evaluation (Yan et al., 2024). The AST evaluation emphasizes the syntactic precision of the generated function calls, ensuring that the model's output adheres to a predefined function documentation in terms of structure and parameters. This includes verifying the correctness of function names, required parameters, and appropriate data types. In contrast, Executable Function Evaluation takes this further by executing the generated function calls to assess their functional accuracy. This evaluation ensures that the functions not only compile but also run correctly, producing the intended outputs, which is vital for real-world applications. In addition to BFCL, we incorporated F1 scores to measure exact matches of API names and parameters in order to evaluate the models on alternative benchmarks (Abdelaziz et al., 2024). In these scenarios, APIs are highly specific, and execution is only feasible if all aspects, such as names, parameters, and input/output formats, strictly conform to the API specifications.

### 5.2 OVERALL PERFORMANCE ON VARIOUS BENCHMARKS

We first evaluate Hammer series on BFCL. Table 2 indicates that within the BFCL framework, our Hammer series consistently achieves corresponding sota performance at comparable scales, particularly Hammer-7B, whose overall performance ranks second only to the proprietary GPT-4. In addition, we evaluated our Hammer series (1.5b, 4b, 7b) on other academic benchmarks to further show our model's generalization ability. Upon observing Hammer's performance across various

Table 4: Detailed performance comparison of different models using Abstract Syntax Tree (AST) evaluation with regard to four function-calling styles on BFCL (as of date 09/20/2024).

| Model | AST Summary | Simple | Multiple | Parallel | Parallel Multiple |
|---|---|---|---|---|---|
| GPT-4-0125-Preview (Prompt) | 85.50 | 78.82 | 88.44 | 91.00 | 83.75 |
| GPT-4-1106-Preview (Prompt) | **86.31** | 78.75 | **89.12** | **94.12** | 83.25 |
| GPT-4-0613 (Prompt) | 84.66 | 78.76 | 85.46 | 91.75 | 82.67 |
| Hammer-7B (FC) | 78.70 | 69.31 | 82.52 | 78.88 | **84.08** |
| GPT-4-turbo-2024-04-09 (Prompt) | 85.41 | **80.47** | 88.81 | 88.12 | 84.25 |
| GPT-4o-mini-2024-07-18 (Prompt) | 80.52 | 75.88 | 81.64 | 85.12 | 79.42 |
| Functionary-Medium-v3.1-70B (FC) | 81.06 | 74.34 | 87.59 | 81.62 | 80.67 |
| Functionary-Small-v3.1-8B (FC) | 78.64 | 72.70 | 83.31 | 85.62 | 72.92 |
| xLAM-7B-fc (FC) | 72.77 | 70.28 | 78.18 | 74.12 | 68.50 |
| Gorilla-OpenFunctions-v2-7B (FC) | 73.18 | 70.81 | 79.47 | 75.75 | 66.67 |
| Functionary-Small-v3.2-8B (FC) | 76.16 | 69.50 | 81.50 | 80.12 | 73.50 |
| FireFunction-v2-70B (FC) | 74.20 | 74.11 | 81.49 | 73.62 | 67.58 |
| Granite-20B-FunctionCalling (FC) | 66.73 | 65.27 | 73.05 | 60.75 | 67.83 |
| Hammer-4B (FC) | 69.59 | 62.58 | 77.72 | 69.12 | 68.92 |
| xLAM-1.3B-fc (FC) | 67.37 | 64.49 | 73.06 | 64.00 | 67.92 |
| Hermes-2-Pro-Llama-3-70B (FC) | 72.09 | 66.29 | 73.49 | 70.25 | 78.33 |
| Hammer-1.5B (FC) | 65.53 | 62.34 | 72.84 | 58.75 | 68.17 |
| Command-R-Plus (FC) | 66.32 | 64.25 | 72.45 | 66.25 | 62.33 |
| Hermes-2-Pro-Llama-3-8B (FC) | 64.18 | 62.32 | 74.96 | 61.62 | 57.83 |
| Hermes-2-Pro-Mistral-7B (FC) | 60.82 | 60.98 | 71.49 | 60.38 | 50.42 |
| Hermes-2-Theta-Llama-3-8B (FC) | 61.08 | 58.53 | 67.82 | 59.62 | 58.33 |

benchmarks unrelated to the xlam-function-calling-60k Datasets, as shown in Table 3, we find that Hammer demonstrates remarkably stable performance, which indicates the robustness of Hammers. Considering that the demand for on-device applications is the primary motivation behind our research, Appendix H further illustrates the non-functional metrics and hardware configurations of our Hammer-7B model when deployed on mobile devices.

## 5.3 DETAILED PERFORMANCE ON DIFFERENT TYPES OF FUNCTION CALLING

In this section, we closely examine the performance of Hammer across different types of function-calling tasks, as exampled in Figure 4, and detailed in Appendix F.

| Simple | Multiple | Parallel | Parallel Multiple | Irrelevance |
|---|---|---|---|---|
| **Query** What is 1 + 2 ? | **Query** What is 1 + 2 ? | **Query** What is (1 + 2) and (3 + 4)? | **Query** What is (1 + 2) and (3 x 4)? | **Query** What is 1 + 2 ? |
| **Candidates** [add(int a, int b)] | **Candidates** [add(int a, int b), mult(int a, int b)] | **Candidates:** [add(int a, int b)] | **Candidates:** [add(int a, int b), mult(int a, int b)] | **Candidates:** [mult(int a, int b)] |
| **Currect Answer** [add(a=1, b=2)] | **Currect Answer** [add(a=1, b=2)] | **Currect Answer** [add(a=1, b=2), add(a=3, b=4)] | **Currect Answer** [add(a=1, b=2), mult(a=3, b=4)] | **Currect Answer** [] |

Figure 4: Demonstration of different function-calling tasks.

As shown in Table 4 and Table 5, we found that Hammer-7B demonstrates exceptional overall performance across these various tasks. Its AST Summary is second only to the GPT-4 series and Functionary-Medium-v3.1-70B. Notably, Hammer-7B even outperformed GPT-4 in the practically relevant Executable Function Evaluation, highlighting the potential of Hammer and the function-masking technique in real-world scenarios. Moreover, we observed that Hammer-7B achieved state-of-the-art results in both tables for the most complex Parallel Multiple task. This suggests that the function-masking training approach becomes increasingly advantageous as task complexity rises. This aligns with our insight that more complex tasks typically demand a deeper understanding of functions, necessitating models to focus more on function descriptions.

## 5.4 ABLATION ON DIFFERENT BASE MODELS

To further validate the effectiveness of our augmented data and tuning technique, we applied our approach to two different sizes of the deepseek-coder models, in addition to the Qwen series. The results are illustrated in Table 6. Upon examining the results presented in the table, we first note that the fine-tuned Hammer model exhibits a notable performance improvement compared to the vanilla Qwen models (Bai et al., 2023; Yang et al., 2024) as well as Qwen variants trained with the original xlam-function-calling-60k dataset, thereby confirming the efficacy of our data and methodology on the Qwen architecture. Subsequently, we compared the deepseek-coder model (Guo et al., 2024)

Table 5: Detailed performance comparison of different models using Executable Function (Exec.) evaluation with regard to four function-calling styles on BFCL (as of date 09/20/2024).

| Model | Exec. Summary | Simple | Multiple | Parallel | Parallel Multiple |
|---|---|---|---|---|---|
| GPT-4-0125-Preview (Prompt) | 89.25 | **99.00** | **96.00** | 82.00 | 80.00 |
| GPT-4-1106-Preview (Prompt) | 87.38 | **99.00** | **96.00** | 82.00 | 72.50 |
| GPT-4-0613 (Prompt) | 87.57 | 98.29 | **96.00** | 86.00 | 70.00 |
| Hammer-7B (FC) | **89.72** | 91.86 | 94.00 | 88.00 | **85.00** |
| GPT-4-turbo-2024-04-09 (Prompt) | 88.13 | **99.00** | **96.00** | 80.00 | 77.50 |
| GPT-4o-mini-2024-07-18 (Prompt) | 87.95 | 98.29 | 94.00 | 82.00 | 77.50 |
| Functionary-Medium-v3.1-70B (FC) | 89.32 | 98.29 | 94.00 | **90.00** | 75.00 |
| Functionary-Small-v3.1-8B (FC) | 83.45 | 87.79 | 90.00 | 86.00 | 70.00 |
| xLAM-7B-fc (FC) | 85.68 | 94.21 | 88.00 | 88.00 | 72.50 |
| Gorilla-OpenFunctions-v2-7B (FC) | 84.97 | 95.86 | **96.00** | 78.00 | 70.00 |
| Functionary-Small-v3.2-8B (FC) | 83.04 | 90.64 | 88.00 | 86.00 | 67.50 |
| FireFunction-v2-70B (FC) | 84.23 | 94.43 | 88.00 | 82.00 | 72.50 |
| Granite-20B-FunctionCalling (FC) | 82.97 | 85.36 | 90.00 | 84.00 | 72.50 |
| Hammer-4B (FC) | 80.82 | 67.79 | 92.00 | 86.00 | 77.50 |
| xLAM-1.3B-fc (FC) | 80.80 | 79.21 | 88.00 | 86.00 | 70.00 |
| Hermes-2-Pro-Llama-3-70B (FC) | 81.29 | 80.64 | 88.00 | 84.00 | 72.50 |
| Hammer-1.5B (FC) | 75.86 | 49.93 | 92.00 | 84.00 | 77.50 |
| Command-R-Plus (FC) | 77.41 | 89.14 | 86.00 | 82.00 | 52.50 |
| Hermes-2-Pro-Llama-3-8B (FC) | 74.05 | 68.71 | 90.00 | 80.00 | 57.50 |
| Hermes-2-Pro-Mistral-7B (FC) | 74.25 | 60.50 | 90.00 | 84.00 | 62.50 |
| Hermes-2-Theta-Llama-3-8B (FC) | 72.54 | 69.14 | 88.00 | 78.00 | 55.00 |

before and after fine-tuning; the fine-tuned variant, referred to as deepseek-coder-Hammer, demonstrates significant enhancements over the vanilla model, despite the poor performance of deepseek-coder prior to fine-tuning. This suggests that our methodology is not exclusively applicable to the Qwen model. Furthermore, it is noteworthy that the performance of the deepseek-coder-Hammer, fine-tuned using our approach, significantly surpasses that of the xLAM model, which was also based on deepseek-coder-instruct and obtained through SFT with the xlam-function-calling-60k dataset. This further underscores the superiority of our proposed method.

Table 6: Ablation on different base models and benchmarks. We apply the masking tuning process to Deepseek-Coder models. Besides, the Qwen-xLAM series are trained with the original xlam-function-calling-60k without masking. For more details see Table 12 and Table 13 in Appendix E.

| Model | API-Bank L-1 | API-Bank L-2 | Tool-Alpaca | Seal-Tools (Single-Tool) | Nexus Raven | F1 Average |
|---|---|---|---|---|---|---|
| Qwen2-7B-Instruct | 60.62 | 49.50 | 48.11 | 77.51 | 63.47 | 59.84 |
| Qwen2-7B-xLAM | 83.38 | 66.14 | 58.18 | 90.98 | 72.74 | 74.48 |
| Hammer-7B | 85.79 | 66.40 | 59.86 | 91.66 | 77.35 | 76.21 |
| Qwen1.5-4B-Chat | 59.78 | 38.48 | 16.98 | 62.32 | 33.70 | 42.25 |
| Qwen1.5-4B-xLAM | 78.64 | 58.32 | 55.48 | 88.28 | 64.10 | 68.96 |
| Hammer-4B | 81.46 | 61.01 | 56.96 | 92.45 | 64.89 | 71.35 |
| Qwen2-1.5B-Instruct | 63.55 | 33.62 | 45.25 | 75.49 | 45.46 | 52.67 |
| Qwen2-1.5B-xLAM | 70.93 | 59.84 | 53.83 | 84.10 | 54.90 | 64.72 |
| Hammer-1.5B | 72.30 | 59.71 | 53.48 | 88.65 | 56.88 | 66.20 |
| xLAM-7B-fc (FC) | 80.69 | 64.24 | 58.96 | 76.87 | 57.50 | 67.65 |
| Deepseek-Coder-7B-Instruct | 56.70 | 39.64 | 20.08 | 65.47 | 46.47 | 45.67 |
| Deepseek-Coder-7B-Hammer | 75.18 | 60.04 | 62.95 | 93.20 | 83.35 | 74.94 |
| xLAM-1.3B-fc (FC) | 83.70 | 64.32 | 50.58 | 80.43 | 54.80 | 66.77 |
| Deepseek-Coder-1.3B-Instruct | 38.42 | 24.75 | 06.06 | 21.52 | 8.15 | 19.78 |
| Deepseek-Coder-1.3B-Hammer | 77.22 | 65.97 | 57.68 | 88.98 | 64.68 | 70.91 |

## 5.5 ABLATION ON DIFFERENT MASKING RATIO

To further investigate the impact of various function masking ratios on model performance, we designed an ablation study focused on the masking ratio. We systematically applied different masking ratios while fine-tuning the Qwen2-1.5B model on the Seal-Tools training dataset for one epoch. Subsequently, we evaluated the performance of the models trained with different masking ratios on the test sets of both Seal-Tools and API-Bank. This allowed us to observe and analyze the performance across both same-task and cross-task scenarios.

Based on the results presented in Figure 5, we observe that an excessively large mask ratio can impede the model's learning speed within the same task scenario, i.e. the test on Seal-Tools. Conversely, the testing results on API-Bank indicate that a larger mask ratio facilitates better generalization of the model across different scenarios. This observation aligns with our previous insights, suggesting that, in the absence of masking, the model may overfit to the training data during fine-tuning, negatively impacting its performance in novel task environments. By enforcing a focus on

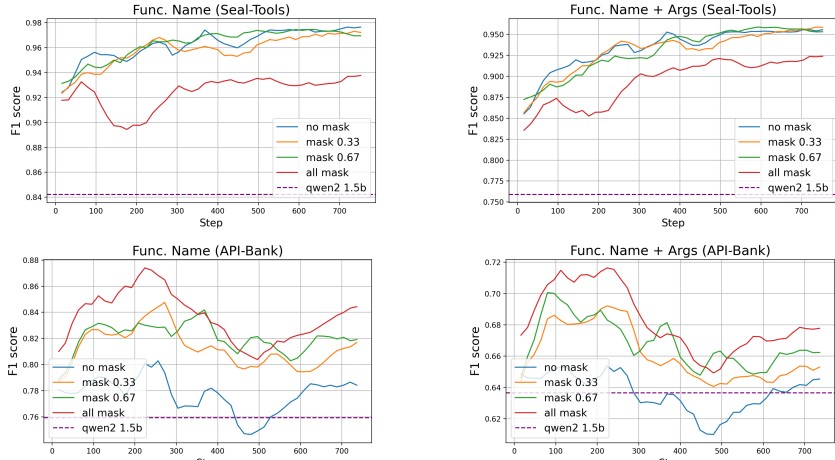

Figure 5: An ablation to evaluate the impact of different masking ratios. For instance, "mask 0.33" denotes that 33% of the instances in the training batch are masked, while others remain unaltered.

more flexible description content, function masking can mitigate this overfitting to some extent, thereby enhancing cross-scenario generalization performance.

## 5.6 ABLATION ON DIFFERENT PROPORTIONS OF IRRELEVANCE-AUGMENTED DATA

To further explore the relationship between these two aspects, we conducted an ablation study on the ratio of irrelevance-augmented data used during training. In this ablation, we sampled a total of 10,000 instances with varying data proportions from the augmented dataset to fine-tune the Qwen2-1.5B-Instruct model and then exam on the BFCL benchmark, observing the changes in the model's irrelevance detection and function-calling capabilities across different ratios.

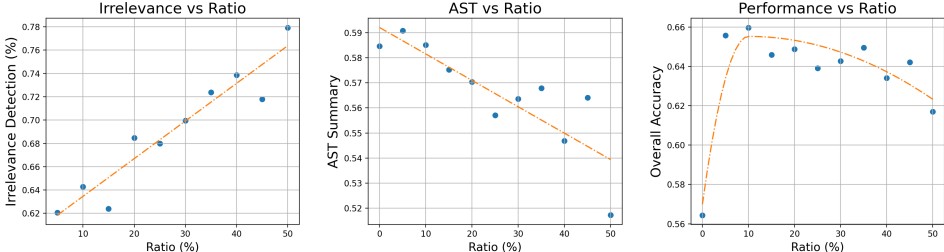

Figure 6: Ablation on different proportions of irrelevance-augmented data applied, e.g. ratio=30% means 30 percent of the training data is sampled from the irrelevance-augmented dataset with other 70 percent sampled from the xlam-function-calling-60k dataset.

As illustrated in the first two panels of Figure 6, the variation in the proportion of irrelevance-augmented data reveals an inverse relationship between the model's performance in irrelevance detection and its function-calling capabilities. This finding underscores the importance of balancing the trade-off between these two aspects. Furthermore, the final panel of Figure 6 indicates that, within our experimental settings, the Hammer model achieves optimal overall performance when the proportion of irrelevance-augmented data is approximately 10%. This insight guides us in establishing the target size for the irrelevance-augmented dataset, i.e., 7.5k. It is essential to note that this proportion may require adjustment depending on the underlying model and training dataset; thus, the ratios presented herein are intended as a reference only.

## 6 CONCLUSION

In conclusion, our exploration of function-calling models reveals significant challenges related to performance inconsistency across different benchmarks, primarily driven by misleading from specific naming conventions. By introducing the Hammer family of models, we provide a robust solution that enhances generalization capabilities through a carefully constructed augmented dataset and innovative function masking techniques. The superior performance of Hammer on a variety of benchmarks demonstrates its potential for practical application in real-world scenarios.

## ACKNOWLEDGEMENTS

The SJTU team is partially supported by the National Key R&D Program of China (2022ZD0114804), Shanghai Municipal Science and Technology Major Project (2021SHZDZX0102) and National Natural Science Foundation of China (62322603). Muning Wen is supported by the Wu Wen Jun Honorary Scholarship, AI Institute, Shanghai Jiao Tong University.

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

## A ANALYSIS OF XLAM MODEL PERFORMANCE ON SEAL-TOOLS AND NEXUS RAVEN BENCHMARKS

This section provides a comprehensive analysis of the factors contributing to the performance degradation of the xLAM model on the Seal-Tools and Nexus Raven benchmarks. Through a detailed examination of failure cases, specific areas where the model encountered difficulties have been identified.

Errors were categorized into three primary types: function selection errors, parameter filling errors, and rejection errors. Table 7 compares the number of cases for each error type, contrasting the Hammer-7b with the xLAM-7b-fc-r.

Table 7: Error type analysis for Seal-Tools and Nexus Raven benchmarks (Hammer-7b | xLAM-7b-fc-r).

| Benchmarks | Function Error | Parameter Error | Rejection Error | Correct | Total |
|---|---|---|---|---|---|
| Seal-Tools | 14 | 7 | 49 | 63 | 3 | 37 | 228 | 187 | 294 | 294 |
| Nexus Raven | 24 | 13 | 65 | 21 | 14 | 186 | 215 | 98 | 318 | 318 |

The data reveal that the xLAM model exhibited a notably high rate of rejection errors, particularly in the Nexus Raven benchmark. This indicates that xLAM may be overly conservative in rejecting candidate functions, resulting in a significant number of missed correct predictions.

Considering the diverse range of parameters in the Nexus Raven benchmark, we further analyzed how the number of parameters in function labels influenced the model's performance.

Table 8: Impact of parameter numbers on performance in the Nexus Raven benchmark (Hammer-7b|xLAM-7b-fc-r).

| Parameter Number | Function Error | Parameter Error | Rejection Error | Correct | Total |
|---|---|---|---|---|---|
| 0 | 0 | 0 | 0 | 0 | 0 | 1 | 6 | 5 | 6 | 6 |
| 1 | 0 | 0 | 7 | 0 | 9 | 15 | 23 | 24 | 39 | 39 |
| 2 | 1 | 3 | 7 | 7 | 0 | 16 | 61 | 43 | 69 | 69 |
| 3 | 0 | 0 | 10 | 3 | 1 | 23 | 26 | 11 | 37 | 37 |
| 4 | 0 | 0 | 4 | 4 | 0 | 14 | 22 | 8 | 26 | 26 |
| 5 | 22 | 10 | 8 | 7 | 0 | 39 | 33 | 7 | 63 | 63 |
| 10 | 1 | 0 | 18 | 0 | 2 | 35 | 14 | 0 | 35 | 35 |
| 27 | 0 | 0 | 11 | 0 | 2 | 43 | 30 | 0 | 43 | 43 |
| All | 24 | 13 | 65 | 21 | 14 | 186 | 215 | 98 | 318 | 318 |

Table 8 indicates that the xLAM model's performance deteriorates significantly when dealing with functions that have five or more parameters. Nearly all such cases experienced incorrect rejections, suggesting that the model struggles with complex functions requiring the management of multiple parameters.

## B ERROR TYPE ANALYSIS

We have conducted categorical statistics for the various failed cases presented in Figure 2 to provide readers with a more detailed and comprehensive understanding. The specific results are shown as Table 9.

In Table 9, "Correct" refers to cases where the prediction is entirely correct; "Func. Error" indicates cases where the function itself was incorrectly selected; "Param. Error" represents cases where the function was correctly chosen, but parameter filling was erroneous; and "Reject Error" denotes cases where the model incorrectly deemed all candidates irrelevant. The table confirms xLAM's overreliance on function and parameter names, as the frequency of various errors increases significantly when function and parameter names are masked.

Table 9: Categorical statistics of number of various failed cases, with "cases with Hammer|cases with xLAM" in each grid.

| Types | No Mask | Fn Mask | Arg Mask | All Mask |
|---|---|---|---|---|
| Correct | 228 \| 187 | 213 \| 174 | 213 \| 148 | 212 \| 133 |
| Func. Error | 14 \| 7 | 20 \| 14 | 20 \| 13 | 23 \| 21 |
| Param. Error | 49 \| 63 | 60 \| 66 | 57 \| 72 | 51 \| 70 |
| Reject Error | 3 \| 37 | 1 \| 38 | 4 \| 61 | 8 \| 70 |
| Total | 294 \| 294 | 294 \| 294 | 294 \| 294 | 294 \| 294 |

## C    DATA CONTAMINATION CLARIFICATION

All queries/examples in the irrelevance-augmented dataset are sampled from the original xlam-function-calling-60k training set. For each sampled example, we simply remove the correct function from its candidate list and replace its label with an empty list, which indicates that all candidates are irrelevant. Therefore, the construction process of the irrelevance-augmented dataset should not introduce any additional data contamination beyond what already exists in the xlam-function-calling-60k dataset. Besides, xlam-function-calling-60k was released in January 2024, while the BFCL-V2 and Seal-Tools benchmarks used in our experiments were released in August and May 2024, respectively, both after the release of the training set. Therefore, Hammer's performance on these two benchmarks should, to some extent, address these concerns.

## D    POTENTIAL FAILURE OF FUNCTION-MASKING

To give readers a better understanding of the limitations of function-masking, in this section, we have to point out that function-masking might perform worse than traditional approaches in the following two cases.

The first case is shown in the ablation study in Section 5.5: *"A large mask ratio can impede the model's learning speed within the same task scenario but facilitate better generalization of the model across different scenarios."* This suggests that when the training and testing sets are from the same source or have similar distributions, function masking might perform worse than traditional approaches.

Another situation arises in cases where there is little description available. In such cases, due to the incomplete information in the descriptions, directly masking all names might be too aggressive, leading to a decline in performance. The comparison on the Tool-Alpaca benchmark suggests this possibility since it contains function candidates in the test set with very limited descriptions and Hammers achieve performance very close to baselines. If the description is further simplified, function masking might perform worse than traditional approaches as well.

# E  EXTRA EXPERIMENTAL RESULTS

Table 10: Full evaluation of the BFCL leaderboard (Yan et al., 2024). (as of date 09/20/2024.)

| Rank | Model | Overall Acc | AST Category | | | | Exec Category | | | | Irrelevance | Relevance |
|---|---|---|---|---|---|---|---|---|---|---|---|---|
| | | | simple | Multiple | Parallel | Parallel Multiple | Simple | Multiple | Parallel | Parallel Multiple | | |
| 1 | GPT-4-0125-Preview (Prompt) | 85.79 | 78.82 | 88.44 | 91.00 | 83.75 | 99.00 | 96.00 | 82.00 | 80.00 | 61.35 | 97.56 |
| 2 | GPT-4-1106-Preview (Prompt) | 85.00 | 78.75 | 89.12 | 94.12 | 83.25 | 99.00 | 96.00 | 72.50 | 64.98 | 90.24 |
| 3 | GPT-4-0613 (Prompt) | 84.74 | 78.76 | 85.46 | 91.75 | 82.67 | 98.29 | 96.00 | 86.00 | 70.00 | 75.57 | 82.93 |
| | Hammer-7b (FC) | 83.92 | 69.31 | 82.52 | 78.88 | 84.08 | 91.86 | 94.00 | 88.00 | 85.00 | 72.87 | 92.68 |
| 4 | GPT-4-turbo-2024-04-09 (Prompt) | 83.89 | 80.47 | 88.81 | 88.12 | 84.25 | 99.00 | 96.00 | 80.00 | 77.50 | 61.82 | 82.93 |
| 5 | GPT-4o-mini-2024-07-18 (Prompt) | 83.35 | 75.88 | 81.64 | 85.12 | 79.42 | 98.29 | 94.00 | 82.00 | 77.50 | 79.20 | 80.49 |
| 6 | GPT-4o-2024-05-13 (Prompt) | 83.13 | 76.18 | 86.01 | 92.12 | 81.00 | 98.00 | 94.00 | 76.00 | 72.50 | 77.44 | 78.05 |
| 7 | Functionary-Medium-v3.1 (FC) | 82.55 | 74.34 | 87.59 | 81.62 | 80.67 | 98.29 | 94.00 | 90.00 | 75.00 | 73.23 | 70.73 |
| 8 | GPT-4-1106-Preview (FC) | 81.78 | 69.32 | 84.19 | 86.38 | 71.92 | 95.43 | 94.00 | 86.00 | 75.00 | 72.70 | 82.93 |
| 9 | Meta-Llama-3-70B-Instruct (Prompt) | 81.59 | 72.87 | 85.91 | 84.00 | 77.83 | 94.14 | 94.00 | 84.00 | 80.00 | 50.47 | 92.68 |
| 10 | Claude-3-Opus-20240229 (Prompt) | 80.88 | 76.65 | 87.47 | 78.38 | 75.17 | 98.57 | 94.00 | 82.00 | 75.00 | 56.15 | 85.37 |
| 11 | GPT-4-0125-Preview (FC) | 80.87 | 68.76 | 84.95 | 80.38 | 74.00 | 84.21 | 94.00 | 88.00 | 75.00 | 74.03 | 85.37 |
| 12 | Nemotron-4-340b-instruct (Prompt) | 80.23 | 68.51 | 80.38 | 78.62 | 79.17 | 86.00 | 90.00 | 80.00 | 77.50 | 84.10 | 78.05 |
| 13 | Functionary-Small-v3.1 (FC) | 80.21 | 72.70 | 83.31 | 85.62 | 72.92 | 87.79 | 90.00 | 86.00 | 70.00 | 68.36 | 85.37 |
| 14 | mistral-large-2407 (FC Any) | 79.66 | 81.01 | 87.42 | 90.50 | 83.50 | 98.29 | 92.00 | 86.00 | 77.50 | 0.34 | 100.00 |
| 15 | GPT-4o-2024-05-13 (FC) | 79.55 | 70.40 | 82.33 | 89.00 | 76.08 | 88.93 | 84.00 | 88.00 | 72.50 | 73.50 | 70.73 |
| 16 | xLAM-7b-fc-r (FC) | 79.41 | 70.28 | 78.18 | 74.12 | 68.50 | 94.21 | 88.00 | 88.00 | 72.50 | 79.76 | 80.49 |
| 17 | GPT-4o-mini-2024-07-18 (FC) | 79.25 | 67.83 | 80.16 | 85.38 | 77.17 | 83.21 | 92.00 | 82.00 | 70.00 | 71.83 | 82.93 |
| 18 | Open-Mixtral-8x22b (Prompt) | 79.14 | 73.47 | 76.14 | 79.12 | 73.67 | 91.86 | 96.00 | 84.00 | 75.00 | 71.42 | 70.73 |
| 19 | Gorilla-OpenFunctions-v2 (FC) | 79.10 | 70.81 | 79.47 | 75.75 | 66.67 | 95.86 | 96.00 | 78.00 | 70.00 | 73.13 | 85.37 |
| 20 | GPT-4-turbo-2024-04-09 (FC) | 79.09 | 64.21 | 82.72 | 82.50 | 75.75 | 88.71 | 88.00 | 86.00 | 72.50 | 79.79 | 70.73 |
| 21 | Functionary-Small-v3.2 (FC) | 78.96 | 69.50 | 81.50 | 80.12 | 73.50 | 90.64 | 88.00 | 86.00 | 67.50 | 72.32 | 80.49 |
| 22 | GPT-4o-2024-08-06 (FC) | 78.87 | 70.71 | 80.97 | 83.25 | 75.58 | 85.36 | 90.00 | 84.00 | 72.50 | 82.91 | 63.41 |
| 23 | mistral-large-2407 (FC Auto) | 78.78 | 68.28 | 86.44 | 90.25 | 83.50 | 76.86 | 92.00 | 86.00 | 77.50 | 48.93 | 78.05 |
| 24 | Claude-3-Sonnet-20240229 (Prompt) | 77.92 | 71.80 | 85.26 | 82.75 | 73.92 | 96.14 | 90.00 | 84.00 | 77.50 | 30.01 | 87.80 |
| 25 | FireFunction-v2 (FC) | 77.45 | 74.11 | 81.49 | 73.62 | 67.58 | 94.43 | 88.00 | 82.00 | 72.50 | 52.94 | 87.80 |
| 26 | Granite-20b-FunctionCalling (FC) | 76.63 | 65.27 | 73.05 | 60.75 | 67.83 | 85.36 | 90.00 | 84.00 | 72.50 | 72.43 | 95.12 |
| 27 | Open-Mistral-Nemo-2407 (Prompt) | 76.31 | 72.89 | 81.37 | 81.50 | 73.75 | 92.50 | 94.00 | 86.00 | 80.00 | 13.25 | 87.80 |
| 28 | Claude-3.5-Sonnet-20240620 (Prompt) | 76.29 | 76.98 | 80.27 | 72.62 | 65.33 | 98.50 | 92.00 | 70.00 | 72.50 | 83.46 | 51.22 |
| | Hammer-4b (FC) | 76.05 | 62.58 | 77.72 | 69.12 | 68.92 | 67.79 | 92.00 | 86.00 | 77.50 | 68.66 | 90.24 |
| 29 | GPT-3.5-Turbo-0125 (FC) | 75.41 | 69.79 | 83.58 | 71.88 | 68.83 | 95.14 | 88.00 | 86.00 | 57.50 | 35.83 | 97.56 |
| 30 | Open-Mistral-Nemo-2407 (FC Auto) | 74.97 | 64.57 | 79.99 | 80.25 | 74.00 | 91.36 | 86.00 | 86.00 | 62.50 | 59.14 | 65.85 |
| 31 | xLAM-1b-fc-r (FC) | 74.90 | 64.49 | 73.06 | 64.00 | 67.92 | 79.21 | 88.00 | 86.00 | 70.00 | 61.21 | 95.12 |
| 32 | Hermes-2-Pro-Llama-3-70B (FC) | 74.78 | 66.29 | 73.49 | 70.25 | 78.33 | 80.64 | 88.00 | 84.00 | 72.50 | 53.80 | 80.49 |
| 33 | Gemini-1.5-Pro-Preview-0514 (FC) | 74.75 | 56.15 | 78.89 | 82.38 | 65.50 | 75.71 | 88.00 | 84.00 | 75.00 | 83.31 | 58.54 |
| 34 | Claude-2.1 (Prompt) | 74.57 | 68.21 | 78.08 | 74.12 | 66.17 | 94.64 | 88.00 | 64.00 | 62.50 | 74.36 | 75.61 |
| 35 | Gemini-1.5-Pro-Preview-0409 (FC) | 74.56 | 55.08 | 79.43 | 83.12 | 64.75 | 76.00 | 88.00 | 80.00 | 72.50 | 83.27 | 63.41 |
| 36 | GPT-4o-2024-08-06 (Prompt) | 74.12 | 65.76 | 76.86 | 72.12 | 71.67 | 70.57 | 88.00 | 78.00 | 75.00 | 89.56 | 53.66 |
| 37 | Command-R-Plus (Prompt) (Original) | 74.11 | 68.14 | 78.13 | 77.50 | 62.17 | 91.29 | 86.00 | 78.00 | 55.00 | 69.31 | 75.61 |
| 38 | Open-Mistral-Nemo-2407 (FC Any) | 73.12 | 67.98 | 82.46 | 77.38 | 76.08 | 92.07 | 86.00 | 86.00 | 62.50 | 0.72 | 100.00 |
| | Hammer-1.5b (FC) | 73.04 | 62.34 | 72.84 | 58.75 | 68.17 | 49.93 | 92.00 | 84.00 | 77.50 | 72.18 | 92.68 |
| 39 | Mistral-Medium-2312 (Prompt) | 72.19 | 63.77 | 80.22 | 69.12 | 59.25 | 93.43 | 88.00 | 70.00 | 57.50 | 84.54 | 56.10 |
| 40 | Command-R-Plus (FC) (Original) | 72.04 | 64.25 | 72.45 | 66.25 | 62.33 | 89.14 | 86.00 | 82.00 | 52.50 | 52.75 | 92.68 |
| 41 | Gemini-1.5-Flash-Preview-0514 (FC) | 70.75 | 65.80 | 83.26 | 63.87 | 63.50 | 57.93 | 86.00 | 74.00 | 75.00 | 74.69 | 63.41 |
| 42 | DBRX-Instruct (Prompt) | 69.55 | 69.97 | 80.35 | 51.50 | 50.50 | 90.50 | 86.00 | 60.00 | 62.50 | 44.86 | 82.93 |
| 43 | Claude-3.5-Sonnet-20240620 (FC) | 68.88 | 73.95 | 82.09 | 65.38 | 62.75 | 95.36 | 86.00 | 44.00 | 40.00 | 75.91 | 63.41 |
| 44 | GPT-3.5-Turbo-0125 (Prompting) | 66.19 | 59.01 | 67.74 | 65.25 | 48.58 | 44.50 | 86.00 | 78.00 | 55.00 | 69.97 | 87.80 |
| 45 | Hermes-2-Pro-Llama-3-8B (FC) | 66.18 | 62.32 | 74.96 | 61.62 | 57.83 | 68.71 | 90.00 | 80.00 | 57.50 | 55.16 | 53.66 |
| 46 | Hermes-2-Pro-Mistral-7B (FC) | 65.44 | 60.98 | 71.49 | 60.38 | 50.42 | 60.50 | 90.00 | 84.00 | 62.50 | 38.55 | 75.61 |
| 47 | Hermes-2-Theta-Llama-3-8B (FC) | 64.83 | 58.53 | 67.82 | 59.62 | 58.33 | 69.14 | 88.00 | 78.00 | 55.00 | 62.66 | 51.22 |
| 48 | Meta-Llama-3-8B-Instruct (Prompt) | 62.70 | 58.53 | 70.26 | 53.50 | 53.25 | 84.50 | 88.00 | 68.00 | 50.00 | 22.88 | 78.05 |
| 49 | Claude-3-Opus-20240229 (FC tools-2024-04-04) | 61.89 | 69.41 | 79.95 | 39.38 | 27.92 | 84.64 | 86.00 | 52.00 | 30.00 | 76.40 | 73.17 |
| 50 | Open-Mixtral-8x7b (Prompt) | 60.82 | 61.49 | 70.70 | 47.12 | 36.83 | 71.86 | 74.00 | 56.00 | 52.50 | 71.84 | 65.85 |
| 51 | Claude-3-Haiku-20240307 (Prompt) | 60.34 | 74.64 | 84.49 | 51.88 | 45.17 | 89.43 | 94.00 | 32.00 | 27.50 | 18.90 | 85.37 |
| 52 | Open-Mixtral-8x22b (FC Any) | 58.89 | 73.23 | 85.42 | 10.75 | 63.08 | 92.57 | 92.00 | 24.00 | 47.50 | 0.34 | 100.00 |
| 53 | Open-Mixtral-8x22b (FC Auto) | 58.37 | 59.75 | 82.75 | 10.50 | 62.33 | 77.79 | 92.00 | 24.00 | 45.00 | 44.20 | 85.37 |
| 54 | Gemini-1.0-Pro-001 (FC) | 57.81 | 64.90 | 79.40 | 38.12 | 22.25 | 86.14 | 84.00 | 58.00 | 5.00 | 67.13 | 73.17 |
| 55 | Mistral-small-2402 (FC Auto) | 55.36 | 51.90 | 82.00 | 15.62 | 34.33 | 87.57 | 90.00 | 14.00 | 20.00 | 77.67 | 80.49 |
| 56 | Mistral-small-2402 (FC Any) | 52.45 | 65.89 | 84.78 | 15.88 | 36.42 | 94.71 | 90.00 | 14.00 | 22.50 | 0.34 | 100.00 |
| 57 | FireFunction-v1 (FC) | 48.11 | 72.25 | 80.37 | 0.00 | 0.00 | 84.79 | 80.00 | 0.00 | 0.00 | 68.55 | 95.12 |
| 58 | Claude-3-Sonnet-20240229 (FC tools-2024-04-04) | 47.97 | 63.79 | 78.37 | 8.25 | 3.33 | 78.50 | 90.00 | 0.00 | 0.00 | 59.89 | 97.56 |
| 59 | Claude-instant-1.2 (Prompt) | 47.95 | 54.50 | 55.81 | 37.75 | 32.42 | 57.50 | 72.00 | 38.00 | 15.00 | 70.21 | 46.34 |
| 60 | Claude-3-Haiku-20240307 (FC tools-2024-04-04) | 47.03 | 72.74 | 78.95 | 1.00 | 2.33 | 90.64 | 92.00 | 6.00 | 0.00 | 29.08 | 97.56 |
| 61 | GPT-4-0613 (FC) | 45.61 | 56.33 | 86.36 | 0.00 | 0.00 | 69.21 | 90.00 | 0.00 | 0.00 | 80.99 | 73.17 |
| 62 | Snowflake/snowflake-arctic-instruct (Prompt) | 42.46 | 34.97 | 31.79 | 42.00 | 38.33 | 33.29 | 28.00 | 60.00 | 40.00 | 65.01 | 51.22 |
| 63 | mistral-large-2407 (Prompt) | 27.87 | 18.08 | 43.11 | 33.38 | 23.17 | 8.71 | 30.00 | 18.00 | 5.00 | 40.70 | 58.54 |
| 64 | Mistral-Small-2402 (Prompt) | 24.44 | 7.83 | 38.97 | 17.25 | 8.92 | 8.14 | 12.00 | 12.00 | 0.00 | 83.22 | 56.10 |
| 65 | Mistral-tiny-2312 (Prompt) | 21.17 | 21.11 | 25.92 | 9.75 | 3.50 | 19.64 | 8.00 | 12.00 | 0.00 | 92.23 | 19.51 |
| 66 | Deepseek-v1.5 (Prompt) | 11.18 | 4.07 | 0.00 | 1.00 | 2.83 | 0.00 | 0.00 | 4.00 | 0.00 | 99.89 | 0.00 |
| 67 | Gemma-7b-it (Prompt) | 10.30 | 2.40 | 0.99 | 0.50 | 0.50 | 1.71 | 0.00 | 0.00 | 0.00 | 96.95 | 0.00 |
| 68 | Hermes-2-Theta-Llama-3-70B (FC) | 10.00 | 0.00 | 0.00 | 0.00 | 0.00 | 0.00 | 0.00 | 0.00 | 0.00 | 100.00 | 0.00 |

Table 11: Full evaluation of different models on several academic benchmarks. The rank is based on the average F1 score on "Func. + Args".

| Model | API-Bank L-1 | API-Bank L-2 | Tool-Alpaca | Seal-Tools (Single-Tool) | Nexus Raven | Func Name | Func.+ Args |
|---|---|---|---|---|---|---|---|
| | | | | | | | |
| GPT-4-0613 (Prompt) | 92.93 \| 84.78 | 69.60 \| 56.98 | 88.64 \| 66.67 | 94.56 \| 93.95 | 95.73 \| 91.60 | 88.29 | 78.79 |
| GPT-4o-mini (Prompt) | 95.08 \| 89.28 | 84.35 \| 67.52 | 64.34 \| 54.69 | 87.94 \| 86.00 | 91.72 \| 84.59 | 84.69 | 76.42 |
| Hammer-7B (FC) | 93.48 \| 85.79 | 82.91 \| 66.40 | 82.31 \| 59.86 | 97.44 \| 91.66 | 92.46 \| 77.35 | 89.72 | 76.21 |
| Granite-20B-FunctionCalling (FC) | 90.41 \| 77.82 | 78.95 \| 59.15 | 77.27 \| 58.00 | 94.86 \| 92.70 | 94.47 \| 75.14 | 87.19 | 72.56 |
| Hammer-4B (FC) | 91.65 \| 81.46 | 77.59 \| 61.01 | 85.09 \| 56.96 | 96.42 \| 92.45 | 81.73 \| 64.89 | 86.50 | 71.35 |
| xLAM-7B-fc (FC) | 90.05 \| 80.69 | 72.49 \| 64.24 | 67.26 \| 58.96 | 78.97 \| 76.87 | 54.09 \| 57.50 | 72.57 | 67.65 |
| Gorilla-OpenFunctions-v2-7B (FC) | 69.21 \| 70.34 | 48.82 \| 54.69 | 72.93 \| 51.26 | 93.20 \| 91.11 | 72.84 \| 68.41 | 71.40 | 67.16 |
| xLAM-1.3B-fc (FC) | 94.86 \| 83.70 | 91.80 \| 64.32 | 64.86 \| 50.58 | 90.74 \| 80.43 | 64.43 \| 54.80 | 81.34 | 66.77 |
| Hammer-1.5B (FC) | 82.13 \| 72.30 | 79.82 \| 59.71 | 80.90 \| 53.48 | 95.59 \| 88.65 | 79.87 \| 56.88 | 83.66 | 66.20 |
| Qwen2-7B-Instruct (Prompt) | 81.55 \| 60.62 | 95.65 \| 49.50 | 71.59 \| 48.11 | 93.88 \| 77.51 | 87.05 \| 63.47 | 85.94 | 59.84 |
| Qwen2-1.5B-Instruct (Prompt) | 74.63 \| 63.55 | 57.69 \| 33.62 | 65.76 \| 45.25 | 82.08 \| 75.49 | 70.62 \| 45.46 | 70.16 | 52.67 |
| Qwen1.5-4B-Chat (Prompt) | 55.33 \| 59.78 | 46.74 \| 38.48 | 35.41 \| 16.98 | 48.44 \| 62.32 | 29.03 \| 33.70 | 42.99 | 42.25 |

Table 12: AST Evaluation for Hammers and different base models on BFCL.

| Overall Acc | Model | AST Summary | Simple | Multiple | Parallel | Parallel Multiple | Irrelevance | Relevance |
|---|---|---|---|---|---|---|---|---|
| 72.79 | Qwen2-7B-Instruct | 69.47 | 68.75 | 81.88 | 60.75 | 66.50 | 61.31 | 97.56 |
| 76.36 | Qwen2-7b-xLAM | 72.16 | 66.55 | 82.05 | 67.38 | 72.67 | 76.84 | 92.68 |
| 80.06 | Hammer-7B | 78.70 | 69.31 | 82.52 | 78.88 | 84.08 | 72.87 | 92.68 |
| 32.92 | Qwen1.5-4B-Chat | 25.43 | 24.60 | 32.99 | 22.12 | 22.00 | 66.56 | 29.27 |
| 71.13 | Qwen1.5-4b-xLAM | 67.75 | 63.88 | 77.84 | 62.62 | 66.67 | 58.20 | 97.56 |
| 72.87 | Hammer-4B | 69.59 | 62.58 | 77.72 | 69.12 | 68.92 | 68.66 | 90.24 |
| 46.90 | Qwen2-1.5B-Instruct | 41.44 | 50.77 | 61.80 | 19.38 | 33.83 | 22.91 | 92.68 |
| 69.42 | Qwen2-1.5b-xLAM | 65.39 | 63.30 | 73.84 | 57.25 | 67.17 | 62.27 | 92.68 |
| 71.16 | Hammer-1.5B | 65.52 | 62.34 | 72.84 | 58.75 | 68.17 | 72.18 | 92.68 |
| | | | | | | | | |
| 75.22 | xLAM-7B-fc (FC) | 72.77 | 70.28 | 78.18 | 74.12 | 68.50 | 79.76 | 80.49 |
| 17.65 | Deepseek-Coder-7B-Instruct | 1.60 | 3.53 | 0.05 | 0.25 | 2.58 | 99.51 | 0.00 |
| 79.09 | Deepseek-Coder-7B-Instruct-Hammer | 76.84 | 71.03 | 84.51 | 77.00 | 74.83 | 67.14 | 100.00 |
| 70.96 | xLAM-1.3B-fc (FC) | 67.37 | 64.49 | 73.06 | 64.00 | 67.92 | 61.21 | 95.12 |
| 16.81 | Deepseek-Coder-1.3B-Instruct | 0.21 | 0.83 | 0.00 | 0.00 | 0.00 | 100.00 | 0.00 |
| 69.71 | Deepseek-Coder-1.3B-Instruct-Hammer | 67.52 | 65.47 | 74.71 | 60.88 | 69.00 | 57.93 | 90.24 |

Table 13: Full evaluation of the ablation study on different base models and benchmarks.

| Model | API-Bank L-1 | API-Bank L-2 | Tool-Alpaca | Seal-Tools (Single-Tool) | Nexus Raven | Func Name | Func.+ Args |
|---|---|---|---|---|---|---|---|
| | | | | | | | |
| Qwen2-7B-Instruct | 81.55 \| 60.62 | 95.65 \| 49.50 | 71.59 \| 48.11 | 93.88 \| 77.51 | 87.05 \| 63.47 | 85.94 | 59.84 |
| Qwen2-7b-xLAM | 93.00 \| 83.38 | 84.01 \| 66.14 | 84.09 \| 58.18 | 95.36 \| 90.98 | 88.46 \| 72.74 | 88.98 | 74.48 |
| Hammer-7B | 93.48 \| 85.79 | 82.91 \| 66.40 | 82.31 \| 59.86 | 97.44 \| 91.66 | 92.46 \| 77.35 | 89.72 | 76.21 |
| Qwen1.5-4B-Chat | 55.33 \| 59.78 | 46.74 \| 38.48 | 35.41 \| 16.98 | 48.44 \| 62.32 | 29.03 \| 33.70 | 42.99 | 42.25 |
| Qwen1.5-4b-xLAM | 86.68 \| 78.64 | 73.17 \| 58.32 | 84.18 \| 55.48 | 93.19 \| 88.28 | 81.28 \| 64.10 | 83.70 | 68.96 |
| Hammer-4B | 91.65 \| 81.46 | 77.59 \| 61.01 | 85.09 \| 56.96 | 96.42 \| 92.45 | 81.73 \| 64.89 | 86.50 | 71.35 |
| Qwen2-1.5B-instruct | 74.63 \| 63.55 | 57.69 \| 33.62 | 65.76 \| 45.25 | 82.08 \| 75.49 | 70.62 \| 45.46 | 70.16 | 52.67 |
| Qwen2-1.5b-xLAM | 81.61 \| 70.93 | 80.11 \| 59.84 | 81.84 \| 53.83 | 92.77 \| 84.10 | 76.58 \| 54.90 | 82.58 | 64.72 |
| Hammer-1.5B | 82.13 \| 72.30 | 79.82 \| 59.71 | 80.90 \| 53.48 | 95.59 \| 88.65 | 79.87 \| 56.88 | 83.66 | 66.20 |
| | | | | | | | |
| xLAM-7B-fc (FC) | 90.05 \| 80.69 | 72.49 \| 64.24 | 67.26 \| 58.96 | 78.97 \| 76.87 | 54.09 \| 57.50 | 72.57 | 67.65 |
| Deepseek-Coder-7B-Instruct | 51.42 \| 56.70 | 35.51 \| 39.64 | 11.58 \| 20.08 | 50.00 \| 65.47 | 26.89 \| 46.47 | 35.08 | 45.67 |
| Deepseek-Coder-7B-Hammer | 83.47 \| 75.18 | 69.17 \| 60.04 | 83.77 \| 62.95 | 96.95 \| 93.20 | 93.75 \| 83.35 | 85.42 | 74.94 |
| xLAM-1.3B-fc (FC) | 94.86 \| 83.70 | 91.80 \| 64.32 | 64.86 \| 50.58 | 90.74 \| 80.43 | 64.43 \| 54.80 | 81.34 | 66.77 |
| Deepseek-Coder-1.3B-Instruct | 35.23 \| 38.42 | 20.41 \| 24.75 | 10.68 \| 06.06 | 16.46 \| 21.52 | 4.34 \| 8.15 | 17.42 | 19.78 |
| Deepseek-Coder-1.3B-Hammer | 85.51 \| 77.22 | 77.68 \| 65.97 | 82.26 \| 57.68 | 95.74 \| 88.98 | 81.37 \| 64.68 | 84.51 | 70.91 |

## F  DIFFERENT TYPES OF FUNCTION-CALLING TASKS

**Simple:** This query style includes straightforward scenarios where a single function call is made based on the user's input with a single provided JSON format API description.

**Multiple:** In this style, user queries could be answered by one of several function calls. The challenge lies in selecting the most appropriate function from multiple provided APIs. It represents one of the most common real-world use cases.

**Parallel:** This query style requires executing multiple function calls simultaneously in response to a single user query, which may consist of one or more sentences but with only one API provided.

**Parallel Multiple:** This query style combines the parallel and multiple categories, where multiple function and API documents are provided, and each function call might be invoked multiple times based on the query's requirements.

**Irrelevance:**  In this query style, no suitable function exists within the candidate options to fulfill users' intent, thus the model should have the ability to detect it and decline the task, rather than making incorrect attempts.

| Simple | Multiple | Parallel | Parallel Multiple | Irrelevance |
|---|---|---|---|---|
| **Query** | **Query** | **Query** | **Query** | **Query** |
| What is 1 + 2 ? | What is 1 + 2 ? | What is (1 + 2) and (3 + 4)? | What is (1 + 2) and (3 x 4)? | What is 1 + 2 ? |
| **Candidates** | **Candidates** | **Candidates:** | **Candidates:** | **Candidates:** |
| [add(int a, int b)] | [add(int a, int b), mult(int a, int b)] | [add(int a, int b)] | [add(int a, int b), mult(int a, int b)] | [mult(int a, int b)] |
| **Current Answer** | **Current Answer** | **Current Answer** | **Current Answer** | **Current Answer** |
| [add(a=1, b=2)] | [add(a=1, b=2)] | [add(a=1, b=2), add(a=3, b=4)] | [add(a=1, b=2), mult(a=3, b=4)] | [] |

Figure 7: Demonstration of different function-calling styles.

## G  IMPACT OF IRRELEVANCE-AUGMENTED DATA ON MODEL PERFORMANCE

We conducted a thorough analysis to determine the number of failed cases resulting from irrelevant function calls, particularly before and after incorporating the irrelevance-augmented dataset into the training regimen for the Hammer model. The analysis provides insights into the model's ability to accurately detect irrelevant function calls.

Table 14: Performance of Hammer-7B "with|without" irrelevance-augmented dataset, where GT stands for "ground truth".

|  | GT Irrelevant | GT Relevant |
|---|---|---|
| Model Predict Irrelevant | 745\|16 | 35\|0 |
| Model Predict Relevant | 370\|1099 | 2450\|2480 |

Table 14 clearly demonstrate the significant improvement in irrelevance detection when the Hammer model is trained with the irrelevance-augmented dataset. Without this augmentation, the model's performance in detecting irrelevant function calls is poor, with almost all ground truth irrelevant cases being predicted as relevant. This analysis underscores the urgency and importance of incorporating irrelevance-augmented data into the training process, which aligns with our motivation for creating such a dataset and highlights its impact on model performance.

## H  ON-DEVICE FUNCTION CALLING PERFORMANCE AND EFFICIENCY

Our research focuses on the development of compact models (no larger than 7B) that achieve a balance between robustness and performance, with a specific emphasis on maintaining accuracy

in function calling—a critical factor for effective task execution on mobile devices. This section provides a comprehensive analysis of the efficiency and performance of our model in on-device function calling scenarios, aimed at meeting the requirements for lightweight, robust models suitable for mobile applications, such as personal assistants. Table 15 presents a detailed evaluation of non-functional metrics and hardware configurations for our Hammer-7B model, after it has been quantized using the Q4_K_M method.

Table 15: Performance metrics of the Hammer-7B model on mobile devices after quantization.

| Model | Quantization Method | Precision | Prefill Speed | Decode Speed | Backend | RAM Usage | Processor | Mobile Model |
|-------|--------------------|-----------|---------------|--------------|---------|-----------|-----------|--------------|
| Hammer-7B | Q4_K_M | 4 bits | 9.4 tokens/sec | 7.9 tokens/sec | Android | 4.7 GB / 16 GB | Snapdragon 8 Gen 3 | OPPO Find X7 Ultra |

Quantization using the Q4_K_M method results in approximately a 2% performance degradation, which remains within an acceptable range, preserving the model's suitability for mobile deployment. These metrics demonstrate that the Hammer-7B model can operate efficiently on mobile devices. However, its inference speed still has significant room for improvement due to the current lack of universal and efficient edge-side inference frameworks. Nevertheless, we are confident that our community will address this issue in the near future.

## I    EXAMPLE INPUT TO MODELS WITH FUNCTION MASKING

The prompted inputs to models in our experiment are exampled as:

```
[BEGIN OF TASK INSTRUCTION]
You are a tool calling assistant. In order to complete the user's
request, you need to select one or more appropriate tools from the
following tools and fill in the correct values for the tool parameters.
Your specific tasks are:
1. Make one or more function/tool calls to meet the request based
   on the question.
2. If none of the function can be used, point it out and refuse to
   answer.
3. If the given question lacks the parameters required by  the function,
   also point it out.
[END OF TASK INSTRUCTION]

[BEGIN OF AVAILABLE TOOLS]
[
    {
        "name": "LxOm64zLyg",
        "description": "Gets hourly weather forecast information for
            given geographical coordinates using the RapidAPI service.",
        "parameters": {
            "TDpjPd": {
                "description": "The latitude of the geographical location.",
                "type\": "int",
                "default": 46.95828
            },
            "78th2U3lFj": {
                "description": "The longitude of the geographical location.",
                "type": "int",
                "default": 10.87152
            }
        }
    },
    {
        "name": "WoDdNSe7e7K5",
        "description": "Fetches weather updates for a given city
```

```
            using the RapidAPI Weather API.",
        "parameters": {
            "LzZsvxUC": {
                "description": "The name of the city for which to
                    retrieve weather information.",
                "type": "str",
                "default": "London"
            }
        }
    },
    {
        "name": "CBrCNmwOERb",
        "description": "Fetches the hourly weather forecast for a
            given location using the RapidAPI service.",
        "parameters": {
            "TDEJ.ZwMt": {
                "description": "The name of the location for which
                    to retrieve the hourly weather forecast.",
                "type": "str",
                "default": "Berlin"
            }
        }
    },
    {
        "name": "1YTQVXkwLY",
        "description": "Returns an air quality forecast for a given
            location.",
        "parameters": {
            "2bkgDA": {
                "description": "The latitude of the location for
                    which the air quality forecast is to be retrieved.",
                "type": "int",
                "default": "35.779"
            },
            "DQi.ReZ16": {
                "description": "The longitude of the location for
                    which the air quality forecast is to be retrieved.",
                "type": "int",
                "default": "-78.638"
            },
            "hF.1": {
                "description": "The number of hours for which the
                    forecast is to be retrieved (default is 72).",
                "type": "int",
                "default": "72"
            }
        }
    }
]

[END OF AVAILABLE TOOLS]

[BEGIN OF FORMAT INSTRUCTION]
The output MUST strictly adhere to the following JSON format, and
NO other text MUST be included.\nThe example format is as follows.
Please make sure the parameter type is correct. If no function
call is needed, please directly output an empty list '[]'
```

[
```

```
    {
        "name": "func_name1",
        "arguments": {
            "argument1": "value1",
            "argument2": "value2"
        }
    },
... (more tool calls as required)
]
```

[END OF FORMAT INSTRUCTION]

[BEGIN OF QUERY]
What are the current weather conditions in Sydney?
[END OF QUERY]

The corresponding output generated by models should be:

```
[
    {
        "name": "WoDdNSe7e7K5",
        "arguments": {
            "LzZsvxUC": "Sydney"
        }
    }
]
```

