# OpenReview forum: "Robust Function-Calling for On-Device Language Model via Function Masking"
_ICLR.cc/2025/Conference — ICLR 2025 Spotlight_

### Official Review · Reviewer_SUbn · 2024-11-03

**Soundness:** 4
**Presentation:** 3
**Contribution:** 3
**Rating:** 8
**Confidence:** 4

**Summary:**

The paper addresses the problem of inconsistent performance across benchmarks in current function-calling models, which often arises from different function and parameter names. The authors developed an augmented dataset and proposed a novel function masking method that guide the model to focus more on function descriptions, instead of names, thereby enhancing its robustness and generalization capabilities across diverse benchmarks.

**Strengths:**

1. This paper identifies a significant challenge in current function calling models that is both practical and impactful, particularly as the function calling models are increasingly deployed in real-world applications where robustness and accuracy are essential.

2. The proposed method is simple but effective, which addresses the problem thoroughly. The method guides the model to focus on function descriptions rather than names. Also, the method is potentially replicable and offers inspiration for ai safety practitioners in other fields who seek to improve model robustness.

3. The evaluations are thorough and effectively demonstrate the method’s applicability to real-world use cases.

4. The authors consider irrelevance detection, which is essential in practical applicaitons, as it reduces the risk of incorrect or unnecessary function calls and enhances reliability.

**Weaknesses:**

The paper claims that the model is designed for on-device function calling; however, it lacks sufficient explanations or evaluation results to demonstrate its efficiency or performance in such scenarios. The authors did not discuss the hardware configurations required to run the model on edge devices or address potential resource constraints. Additionally, the authors did not provide evaluation results regarding inference time, which are essential for assessing the model’s practicality in real-world, resource-limited environments.

**Questions:**

see weakness

---

> ### Author Response · Authors · 2024-11-22
> **Reply to Reviewer SUbn**
>
> ### We thank Reviewer SUbn for his/her recognition of our work and constructive comments that will surely turn our paper into a better shape.
>
> > **Q1** The paper claims that the model is designed for on-device function calling; however, it lacks sufficient explanations or evaluation results to demonstrate its efficiency or performance in such scenarios. The authors did not discuss the hardware configurations required to run the model on edge devices or address potential resource constraints. Additionally, the authors did not provide evaluation results regarding inference time, which are essential for assessing the model’s practicality in real-world, resource-limited environments.
>
> **A1** We apologize for making this confusion. The demand for on-device applications is the primary motivation behind our research. Initially, our goal was to train an effective function-calling model to serve as the foundation for building on-device agents, such as mobile personal assistants. The accuracy of function-calling is critical, as it largely determines the success of task execution for on-device agents. Based on this, we focused on key aspects such as lightweight models (no larger than 7B), robustness, and balancing irrelevance detection with function-calling performance. The table below further illustrates the non-functional metrics and hardware configurations of our Hammer-7B model when deployed on mobile devices after quantization:
>
> | Benchmarks| Quantization Method | Precision |  Prefill Speed  |   Decode Speed  | Backend |  RAM usage  |Processor|
> |:---------:|:-------------------:|:---------:|:---------------:|:---------------:|:-------:|:-----------:|:-----------:|
> |Hammer-7b  |        Q4_K_M       |  4-bits   | 9.4 tokens/sec. | 7.9 tokens/sec. | Android | 4.7GB/16 GB |Snapdragon 8 Gen 3|
>
> After quantization, the model experiences approximately 2% performance degradation, which remains within an acceptable and reasonable range. The table demonstrates that our Hammer-7B model is already capable of running on mobile devices. However, its inference speed still has significant room for improvement due to the current lack of universal and efficient edge-side inference frameworks. Nevertheless, we are confident that our community will address this issue in the near future.
>
> We greatly appreciate the reviewer's reminder. We will include this table in the revision to further clarify the rationale behind our choice of model size and its relevance to on-device applications.

---

### Official Review · Reviewer_95n3 · 2024-11-04

**Soundness:** 2
**Presentation:** 3
**Contribution:** 2
**Rating:** 6
**Confidence:** 4

**Summary:**

This paper introduces Hammer, a novel family of language models specifically designed for function-calling tasks. The authors identify a critical limitation in existing models: their performance varies significantly across benchmarks due to being misled by function and parameter naming conventions. To address this, they propose a function masking technique during training and augment the training data with irrelevant detection examples. Their empirical results show that Hammer models achieve state-of-the-art performance across multiple benchmarks, particularly impressive for their parameter count, with the 7B parameter version competing with much larger models including GPT-4 on certain metrics.

**Strengths:**

1. The problem identification is well-motivated and articulated.
2. The evaluations are comprehensive, covering multiple benchmarks and including detailed ablation studies on both masking ratios and irrelevance detection data proportions.

**Weaknesses:**

1. The paper doesn't include the comparison with some recent relevant baselines (e.g., ToolBench, API-Bank).
2. There are no ablation studies on the effect of different description formats/lengths on model performance.
3. There’s no discussion of potential failure modes or edge cases where function masking might perform worse than traditional approaches.
4. The random string generation process for masking isn't fully specified - different approaches to generating random strings could affect reproducibility.
5. The exploration of the trade-off between masking and maintaining semantic meaning in function names is limited.

**Questions:**

How does the function masking technique affect the model's zero-shot generalization to entirely new API schemas or documentation formats not seen during training?

---

> ### Author Response · Authors · 2024-11-22
> **Reply to Reviewer 95n3: Part One**
>
> ### We thank Reviewer 95n3 for his/her constructive comments that will surely turn our paper into a better shape.
>
> > **Q1** The paper doesn't include the comparison with some recent relevant baselines (e.g., ToolBench, API-Bank).
>
> **A1** We appreciate the reviewer for recommending these relevant works. In our work, API-Bank serves as one of the primary benchmarks, and we have conducted comprehensive experiments based on it, with results presented in Tables 1, 3, 5, and Figure 5, and a detailed introduction provided in Section 5.1.
>
> Regarding ToolBench, we also noticed this benchmark. However, we chose not to utilize it for the following reasons:
>
> 1. Access Limitations: ToolBench requires applying a specific ToolBench key for access. We attempted to request it early on, but unfortunately, we never received a response. We also observed that many other users reflected the same issue at their code repo.
>
> 2. Potential Data Contamination: The xLAM team revealed in their recently released paper [1] that they generate the xLAM-function-calling-60k dataset, which serves as our train set, with ToolBench. This poses a significant risk of data contamination if ToolBench is used as a benchmark, potentially leading to unfair advantages compared with certain baselines.
>
> Nevertheless, based on the latest findings outlined in Point 2, we will update Section 5.1 to include a discussion on ToolBench, providing further clarification on the rationale behind our benchmark selection.
>
> [1] Zhang, J., Lan, T., Zhu, M., Liu, Z., Hoang, T., Kokane, S., ... & Xiong, C. (2024). xlam: A family of large action models to empower ai agent systems. arXiv preprint arXiv:2409.03215.
>
> > **Q2** There are no ablation studies on the effect of different description formats or lengths on model performance.
>
> **A2** We are very grateful for this insightful suggestion. We have conducted an ablation based on the Live dataset in BFCL-V2 to further investigate the effect of different description lengths in labels on model performance (Hammer-7B and xLAM-7b), with results shown as below:
>
> | Desc. Lengths (#tokens) | Hammer Pass Count | xLam Pass Count | Test Case Count | Hammer AST Accuracy | xLam AST Accuracy |  delta  |
> |-----|----|----|-------|------|------|----|
> | (10, 14] | 99 | 81   | 150   | 0.66 | 0.54     | 0.12     |
> | (14, 18] | 717 | 627  | 885   | 0.81 | 0.71     | 0.10     |
> | (18, 22] | 741| 654  | 1062  | 0.70  | 0.62     | 0.08     |
> | (22, 26] | 435| 396  | 636   | 0.68  | 0.62     | 0.06     |
> | (26, 30] | 150 | 132  | 195   | 0.77 | 0.68     | 0.09     |
> | (30, 34] | 21 | 15   | 27    | 0.78 | 0.56     | 0.22     |
> | (34, 38] | 57 | 45   | 96    | 0.59 | 0.46     | 0.13     |
>
> Based on the results in the table, we observe that Hammer demonstrates a greater advantage in cases with longer descriptions, while its advantage is less pronounced in cases with shorter descriptions. However, since the number of samples with long descriptions in the test set is relatively small, the confidence of this conclusion based solely on the table may not be sufficiently robust. Nonetheless, the analysis in A5 provides supplementary evidence: Hammer's performance advantage is lower on Tool-Alpaca, which contains function candidates in the test set with very limited descriptions, compared to other benchmarks. This further corroborates our observation from a different perspective.
>
> > **Q3** There’s no discussion of potential failure modes or edge cases where function masking might perform worse than traditional approaches.
>
> **A3** Thank you very much for the reviewer’s suggestion. The statement that "function masking might perform worse than traditional approaches" could be true in the following two cases.
>
> The first case is shown in the ablation study in our Section 5.5 with result in Figure 5 and summary in lines 486-489: "A large mask ratio can impede the model’s learning speed within the same task scenario but facilitate better generalization of the model across different scenarios." This suggests that when the training and testing sets are from the same source or have similar distributions, function masking might perform worse than traditional approaches.
>
> Another situation arises in cases where there is little description available. In such cases, due to the incomplete information in the descriptions, directly masking all names might be too aggressive, leading to a decline in performance. The comparison on the Tool-Alpaca benchmark suggests this possibility since it contains function candidates in the test set with very limited descriptions and Hammers achieve perormance very close to baselines. If the description is further simplified, function masking might perform worse than traditional approaches as well. Fortunately, we also had this concern during Hammer's training and proposed trade-off referred to **A5**.
>
> Thanks again to the reviewer for their valuable feedback. We will include this discussion in the experimental section of the paper.

---

> > ### Comment · Reviewer_95n3 · 2024-11-22
> >
> > Thank you for your feedback. The responses address most of my concerns. I will raise my score.

---

> > > ### Author Response · Authors · 2024-11-23
> > > **Thank you for the time and effort you dedicated to re-evaluating our work!**
> > >
> > > We would like to say thank you sincerely for your recognition of our efforts. Your constructive feedback has been invaluable in guiding our revisions and enhancing the quality of our work. Thanks a lot!

---

> ### Author Response · Authors · 2024-11-22
> **Reply to Reviewer 95n3: Part Two**
>
> > **Q4** The random string generation process for masking isn't fully specified - different approaches to generating random strings could affect reproducibility.
>
> **A4** We sincerely thank the reviewer for their thoughtful reminder. We will add the following details regarding the randomization process in Section 4.1: *"We first determine the length $L$ of the random string using `random.randint(5, 15)`. Then, we randomly select $L$ characters from a character set comprising 52 uppercase and lowercase letters and 10 digits, to replace the corresponding function or parameter names."*
>
> Additionally, to ensure reproducibility, we have made all relevant code and training data publicly available in the supplementary material.
>
> > **Q5** The exploration of the trade-off between masking and maintaining semantic meaning in function names is limited.
>
> **A5** Thank you very much to the reviewer for raising this concern. We also noticed this during Hammer's training. Considering the complexities of real-world scenarios, we do not apply the masking strategy to all the training samples. Instead, we set a masking ratio (We found that a masking ratio of 0.33 yields the best overall performance for Hammer across all benchmarks). Before the start of each training epoch, we randomly mask a specific ratio of the samples. This approach ensures a balance between allowing the LLM to infer functionality from names and encouraging it to focus on descriptions. The strong performance of Hammer on the Tool-Alpaca benchmark confirms the effectiveness of this trade-off since Tool-Alpaca contains function candidates in the test set with very limited descriptions.
>
> > **Q6** How does the function masking technique affect the model's zero-shot generalization to entirely new API schemas or documentation formats not seen during training?
>
> **A6** Thank you for your thoughtful question. Theoretically, the function masking technique might offer a slight advantage over traditional methods in addressing zero-shot generalization to entirely new API schemas. This is because it at least circumvents the naming style drift introduced by designers' preferences in new API schemas. However, considering the overall formats, the improvement brought by directly applying function masking may be limited. The primary reason lies in the zero-shot setting, where the model may struggle to locate or distinguish among function names, parameter names, and function descriptions due to a lack of structural knowledge about the new schemas.
>
> To address this challenge, we can consider modifying the masking technique. Specifically, by randomizing function names and parameter names while adding type tags (e.g., `param: random_str`, `func: random_str`, and `desc: xxxx`), the model can be trained to locate each element using the tags such as `param:`, `func:`, and `desc:`. Intuitively, this modified masking approach should further enhance the trained model's performance in zero-shot generalization to entirely new API schemas.
>
> We hope this response meets your satisfaction.

---

### Official Review · Reviewer_nBzQ · 2024-11-04

**Soundness:** 3
**Presentation:** 3
**Contribution:** 4
**Rating:** 8
**Confidence:** 3

**Summary:**

In this paper, the authors studied large language models' function calling and tool use abilities. The authors not only identified a critical gap in existing function-calling models, where the model is often misled by specific naming of functions and arguments.
To address the issue, the authors present a novel tunning framework that masks the function name and arguments; this results in a family of various new open-source function calling models called Hammer and also a new specialized dataset for irrelevance detection. These models achieves state of the art performance on open benchmark and leaderboards such as Berkeley function calling (BFCL).

**Strengths:**

- The authors identified a very important issue of the current existing open source function calling models such as xlam.
- Based on the issue, the method that the authors proposed makes a lot of sense and indeed achieves very good performance.
- The model and dataset will be open source later, which would benefit the community a lot.

**Weaknesses:**

- The method is specifically designed for function calling problems; it is unclear whether some other domains will also benefit the tunning framework;
- I am not sure how this strategy is specifically related to on-device language models; It can also be applied to larger models, and it would be really great to see the performance of larger models such as 70B llama if they are applied with this method;
- For the cases where there is little description, I am not sure if we can still leverage the masking strategy.

**Questions:**

Please see the weakness part.

---

> ### Author Response · Authors · 2024-11-22
> **Reply to Reviewer nBzQ: Part One**
>
> ### We thank Reviewer nBzQ for his/her recognition of our work and constructive comments that will surely turn our paper into a better shape.
>
> > **Q1** The method is specifically designed for function calling problems; it is unclear whether some other domains will also benefit the tunning framework.
>
> **A1** We appreciate the reviewer's thoughtful question. While the tuning framework in this work is primarily applied to enhance the model's function-calling performance, we believe its core idea holds potential benefits for other domains as well. Meanwhile, we also extend our gratitude to Reviewer SUbn for his insight that *"this method is potentially replicable and offers inspiration for ai safety practitioners in other fields who seek to improve model robustness."*
>
> Besides, the central insight of this work lies in leveraging a masking mechanism to guide the model's attention away from incomplete, potentially misleading, or redundant input elements and toward more complete and robust components, thereby improving overall model performance. This approach could benefit any task with similar characteristics. For example, when training a web agent, we could mask out website names or domain names to encourage the model to focus on the site's description. Similarly, when training LLMs to play various card games, we might guide the model to focus on the functionality of the cards rather than their names. From another perspective, a well-tuned function-calling model serves as a foundational capability for deploying LLMs across many domains, as the specific actions executed in numerous tasks can often be conceptualized as variations of "function-calls", such as the agent selection in multi-agent communication tasks.
>
> Further, considering that different domains may prioritize different information, in the future we also plan to explore the design of more dynamic masking mechanisms as alternatives to static name masking.
>
> > **Q2** I am not sure how this strategy is specifically related to on-device language models; It can also be applied to larger models, and it would be really great to see the performance of larger models such as 70B llama if they are applied with this method.
>
> **A2** We apologize for making this confusion. The demand for on-device applications is the primary motivation behind our research. Initially, our goal was to train an effective function-calling model to serve as the foundation for building on-device agents, such as mobile personal assistants. The accuracy of function-calling is critical, as it largely determines the success of task execution for on-device agents. Based on this, we focused on key aspects such as lightweight models (no larger than 7B), robustness, and balancing irrelevance detection with function-calling performance. The table below further illustrates the non-functional metrics of our Hammer-7B model when deployed on mobile devices after quantization:
>
> | Benchmarks| Quantization Method | Precision |  Prefill Speed  |   Decode Speed  | Backend |  RAM usage  |Processor|
> |:---------:|:-------------------:|:---------:|:---------------:|:---------------:|:-------:|:-----------:|:-----------:|
> |Hammer-7b  |        Q4_K_M       |  4-bits   | 9.4 tokens/sec. | 7.9 tokens/sec. | Android | 4.7GB/16 GB |Snapdragon 8 Gen 3|
>
> After quantization, the model experiences approximately a 2% performance degradation, which remains within an acceptable and reasonable range. We will include this table in the revison to further clarify the rationale behind our choice of model size and its relevance to on-device applications.
>
> Moreover, we agree that our strategy is applicable to larger models. In fact, we are very eager to explore the performance limits on even larger models, such as 70B, to push the boundaries of what can be achieved. However, due to our limited computational resources, it has been challenging to complete the training of a 70B model within the constraints of a limited time window (we are still actively working on this). At the same time, this limitation is also one of the key motivations for making all our materials—code, data, models, etc.—open source, since we sincerely hope that interested researchers will join us in this exploration.

---

> ### Author Response · Authors · 2024-11-22
> **Reply to Reviewer nBzQ: Part Two**
>
> > **Q3** For the cases where there is little description, I am not sure if we can still leverage the masking strategy.
>
> **A3** Thank you very much to the reviewer for raising this important question. We also have this concern during Hammer's training. Considering the complexities of real-world scenarios, we do not apply the masking strategy to all the training samples. Instead, we set a masking ratio (We found that a masking ratio of 0.33 yields the best overall performance for Hammer across all benchmarks). Before the start of each training epoch, we randomly mask a specific ratio of the samples. This approach ensures a balance between allowing the LLM to infer functionality from names and encouraging it to focus on descriptions. The strong performance of Hammer on the Tool-Alpaca benchmark confirms the effectiveness of this trade-off since Tool-Alpaca contains function candidates in the test set with very limited descriptions.
>
> In addition, in Tool-Alpaca, even the performance achieved by GPT-4 is quite limited. This highlights another insight we aim to convey: in real-world scenarios requiring function calls, providing comprehensive natural language descriptions should be prioritized. That is, in practice, when descriptions are incomplete, supplementing functions in the library with detailed descriptions is often simpler, more direct, and more efficient than training a model to infer the function's purpose from the limited information.
>
> We hope this clarifies our rationale and approach. Thanks again for this valuable feedback.

---

### Official Review · Reviewer_zD7o · 2024-11-04

**Soundness:** 3
**Presentation:** 3
**Contribution:** 3
**Rating:** 6
**Confidence:** 4

**Summary:**

This paper introduces Hammer, a family of lightweight models for function-calling tasks that addresses the problem of inconsistent performance across different benchmarks. The authors propose a function masking technique during training and an irrelevance-augmented dataset to improve model robustness and generalization. Experiments show that Hammer achieves competitive performance compared to larger models like GPT-4 on various benchmarks.

**Strengths:**

1. The function masking technique is an innovative solution to reduce model dependency on naming conventions. The empirical results show this approach helps achieve more consistent performance across different benchmarks.
2. The authors conduct extensive experiments across multiple benchmarks and provide detailed ablation studies on masking ratios and irrelevance detection. The evaluation is thorough and well-documented with clear performance metrics.
3. The work addresses a real-world problem in function-calling models and provides a lightweight solution suitable for on-device deployment. The improved generalization capability has significant practical value.

**Weaknesses:**

1. The authors replace function names with random strings, but don’t explore using semantically similar names. This approach may be overly aggressive since function names often contain valuable semantic information that could be preserved while still improving generalization
2. While the paper demonstrates Hammer’s superior performance, it lacks detailed analysis of other potential contributing factors beyond the masking technique. The choice of base model and other training details could significantly impact the results
3. More detailed error patterns analysis would be beneficial to understand the model and baseline failure patterns across different benchmarks

**Questions:**

1. Have you tried replacing function names with semantically similar ones instead of random strings? This could be more meaningful since function names are important features in practice and descriptions and parameters may sometimes be missing. In some usage scenarios LLMs need to infer functionality from names alone.
2. Could you provide more insights into other factors contributing to Hammer’s performance? Have you tried using DeepSeek models as base models (just like xLAM) instead of just Qwen? This would be helpful to understand what specific aspects of the base model selection impacted results. Also, it would be beneficial to know if there are other training tricks beyond data augmentation.
3. For Table 1’s inconsistent performance across benchmarks, I am curious why does xLAM perform well on three benchmarks but poorly on two others? Could the authors provide some error pattern analysis and related insights?

Minor things: the observations in the paper might be related to the code data contamination problem as well, where the memorization of func names / orders could impact the generalization ability. Ref: https://arxiv.org/pdf/2402.05980

---

> ### Author Response · Authors · 2024-11-22
> **Reply to Reviewer zD7o: Part One**
>
> ### We thank Reviewer zD7o for his/her constructive comments that will surely turn our paper into a better shape.
>
> > **Q1** The authors replace function names with random strings, which may be overly aggressive since function names often contain valuable semantic information. Have you tried replacing function names with semantically similar ones instead of random strings? This could be more meaningful since function names are important features in practice and descriptions and parameters may sometimes be missing. In some usage scenarios LLMs need to infer functionality from names alone.
>
> **A1** Thank you very much to the reviewer for raising this important question.
>
> In fact, our initial approach to enhancing the model's robustness also focused on leveraging semantically similar rewrites with GPT-4o for data augmentation. This approach indeed provided some improvements in robustness. However, we ultimately decided against adopting this strategy for the following reasons:
> 1. The xlam-function-calling-60k dataset already incorporates this aspect by including samples with semantically similar names. For instance, there are multiple function examples for retrieving addresses based on IP:
>     * get_ip_location: Retrieves the latitude and longitude coordinates of a given IP address.
>     * ip_lookup: Fetches the geographical information for a given IP address.
>     * get_place_by_my_ip: Fetches the geographical location related to the IP address of the requester.
>     * get_geolocation_data: Fetches geolocation data for a given IP address.
> 2. The cost of performing GPT-based rewrites for all data is significantly higher than that of random masking operations. Additionally, the pipeline for this approach is more complex. Given the dataset's inherent inclusion of semantically similar samples (as mentioned above), the marginal benefits of using GPT-based rewrites are relatively limited compared with masking operations.
>
> Regarding the concern about overly aggressive issues, we also noticed this during Hammer's training. Considering the complexities of real-world scenarios, we do not apply the masking operatin to all the training samples. Instead, we set a masking ratio (We found that a masking ratio of 0.33 yields the best overall performance for Hammer across all benchmarks). Before the start of each training epoch, we randomly mask a specific ratio of the samples. This approach ensures a balance between allowing the LLM to infer functionality from names and encouraging it to focus on descriptions. The strong performance of Hammer on the Tool-Alpaca benchmark confirms the effectiveness of this trade-off since Tool-Alpaca contains function candidates in the test set with very limited descriptions.
>
> We hope this clarifies our rationale and approach. Thanks again for this valuable feedback.

---

> ### Author Response · Authors · 2024-11-22
> **Reply to Reviewer zD7o: Part Two**
>
> > **Q2** Could you provide more insights into other factors contributing to Hammer’s performance? Have you tried using DeepSeek models as base models (just like xLAM) instead of just Qwen? This would be helpful to understand what specific aspects of the base model selection impacted results. Also, it would be beneficial to know if there are other training tricks beyond data augmentation.
>
> **A2:**
>
> **Comparison of Different Base Models**
>
> As highlighted in Section 5.5 (Table 5), we have experimented with DeepSeek models as base models, similar to xLAM. The results demonstrate that our approach applied to DeepSeek also outperforms xLAM in terms of overall performance. Even more surprisingly—and here we would like to sincerely thank reviewer peJz for their insightful suggestion—upon further investigation of the training details of xLAM models from their newly released paper [1], we discovered that xLAM-fc models might still have 50% of their training data that had not been disclosed in the original paper [2]. This finding indicates that our pipeline achieves superior performance across multiple benchmarks despite utilizing only half of the available data.
>
> **Inspiration from Semantically Similar Cases**
>
> Beyond the masking technique, the reviewer’s point regarding semantically similar cases has inspired us to consider another potential factor contributing to improved performance: further balancing the ratio between semantically similar samples and masked samples. While we have not yet adjusted the ratio of semantically similar data in xLAM-function-calling-60k, we recognize this as a promising avenue for future exploration. We are deeply grateful to the reviewer for this valuable suggestion.
>
> **Future Directions in Data Augmentation**
>
> Finally, apart from data augmentation, there exist approaches like [3,4] that dynamically select different subsets of data for training across epochs to enhance performance. Beyond this method's utility in improving training outcomes, we are also planning to explore a metric-based approach to dynamically determine which data samples should be masked rather than relying on random masking. We believe this could further enhance the performance of our models.
>
> [1] Zhang, J., Lan, T., Zhu, M., Liu, Z., Hoang, T., Kokane, S., ... & Xiong, C. (2024). xlam: A family of large action models to empower ai agent systems. arXiv preprint arXiv:2409.03215.
>
> [2] Liu, Z., Hoang, T., Zhang, J., Zhu, M., Lan, T., Kokane, S., ... & Xiong, C. (2024). Apigen: Automated pipeline for generating verifiable and diverse function-calling datasets. arXiv preprint arXiv:2406.18518.
>
> [3] Yang, Y., Wang, H., Wen, M., & Zhang, W. (2024). P3: A Policy-Driven, Pace-Adaptive, and Diversity-Promoted Framework for Optimizing LLM Training. arXiv e-prints, arXiv-2408.
>
> [4] Wang, X., Zhou, Y., Chen, H., & Zhu, W. (2024, May). Curriculum Learning: Theories, Approaches, Applications, Tools, and Future Directions in the Era of Large Language Models. In Companion Proceedings of the ACM on Web Conference 2024 (pp. 1306-1310).

---

> ### Author Response · Authors · 2024-11-22
> **Reply to Reviewer zD7o: Part Three**
>
> > **Q3** More detailed error patterns analysis would be beneficial to understand the model and baseline failure patterns across different benchmarks. For Table 1’s inconsistent performance across benchmarks, I am curious why does xLAM perform well on three benchmarks but poorly on two others? Could the authors provide some error pattern analysis and related insights?
>
> **A3**
> Thanks for the reviewer's suggestion. We have conducted categorical statistics for the various types of failed cases appeared in the Seal-Tools and Nexus Raven benchmarks where xLAM performs poorly, to provide readers with a more detailed and comprehensive understanding. The specific results are as follows, with "`#cases with Hammer | #cases with xLAM`" in each grid:
>
> | Benchmarks| Func. Error | Param. Error | Reject Error |   Correct  |    Total   |
> |:---------:|:-----------:|:------------:|:------------:|:----------:|:----------:|
> |Seal-Tools |  14 \| 7    |  49 \| 63    |   3 \| 37    | 228 \| 187 | 294 \| 294 |
> |Nexus Raven|  24 \| 13   |  65 \| 21    |  14 \| 186   | 215 \| 98  | 318 \| 318 |
>
> In this context, "Correct" refers to the number of cases where the outputs are entirely correct; "Func. Error" indicates cases where the function itself was incorrectly selected; "Param. Error" represents cases where the function was correctly chosen, but parameter filling was erroneous; and "Reject Error" denotes cases where the model incorrectly deemed all candidates irrelevant.
>
> From the table above, we observe that the performance degradation of xLAM on these two benchmarks primarily stems from an overly aggressive rejection strategy, which fails to accurately assess the relevance between candidates and queries. This issue is particularly pronounced in Nexus Raven. Then, we conducted a further investigation into the test data of Nexus Raven and found that the distribution of parameter number in its function library is exceptionally broad, ranging from 0 to 27 parameters, whereas other benchmarks typically do not exceed 10 parameters. Consequently, we further analyzed the impact of varying numbers of parameters in the labels within Nexus Raven on the performance of xLAM and Hammer. The details are as follows:
>
> |Param. num| Func. Error | Param. Error | Reject Error |   Correct   |    Total   |
> |:--------:|:-----------:|:------------:|:------------:|:-----------:|:----------:|
> |        0 |    0 \| 0   |     0 \| 0   |     0 \| 1   |    6 \| 5   |   6 \| 6   |
> |        1 |    0 \| 0   |     7 \| 0   |     9 \| 15  |   23 \| 24  |  39 \| 39  |
> |        2 |    1 \| 3   |     7 \| 7   |     0 \| 16  |   61 \| 43  |  69 \| 69  |
> |        3 |    0 \| 0   |    10 \| 3   |     1 \| 23  |   26 \| 11  |  37 \| 37  |
> |        4 |    0 \| 0   |     4 \| 4   |     0 \| 14  |   22 \| 8   |  26 \| 26  |
> |        5 |   22 \| 10  |     8 \| 7   |     0 \| 39  |   33 \| 7   |  63 \| 63  |
> |       10 |    1 \| 0   |    18 \| 0   |     2 \| 35  |   14 \| 0   |  35 \| 35  |
> |       27 |    0 \| 0   |    11 \| 0   |     2 \| 43  |   30 \| 0   |  43 \| 43  |
> |      All |   24 \| 13  |    65 \| 21  |    14 \| 186 |  215 \| 98  | 318 \| 318 |
>
> From the table above, we can observe that xLAM's rejection rate significantly increases when handling functions with parameters of five or more, and all such cases are rejected when the number of parameters reaches ten or more. In contrast, the hammer method, which employs a masking technique, is still able to maintain a certain level of accuracy. We sincerely thank the reviewer for their insightful guidance. This analysis indirectly corroborates that xLAM exhibits misleading issues caused by the parameter name when assessing the relevance between candidates and queries (as mentioned in Section 3.1 of our paper).

---

### Official Review · Reviewer_peJz · 2024-11-04

**Soundness:** 3
**Presentation:** 2
**Contribution:** 3
**Rating:** 6
**Confidence:** 4

**Summary:**

This study introduces Hammer, a novel family of foundation models designed to enhance function-calling capabilities in large language models (LLMs). The authors developed a specialized dataset comprising 7,500 instances to improve the models' sensitivity to irrelevant functions. By shifting the focus from function names and parameters to their descriptions, the Hammer models aim to reduce the likelihood of misinterpretation, thereby enhancing generalization across diverse benchmarks. Experimental results indicate that Hammer models achieve superior performance across various benchmarks.

**Strengths:**

1. The work effectively identifies the issue of naming variability in function-calling LLMs and incorporates training methods to address it.
2. Hammer models utilize an augmented dataset to handle null function-calling cases.
3. The study conducts comprehensive evaluations, demonstrating that Hammer consistently performs well across diverse benchmarks, establishing it as a reliable open-source model for function calling.

**Weaknesses:**

1. The proposed methods may inherit potential biases inherent in LLMs. The sampled "Masked Data" can degrade in performance when encountering function names or parameters that differ significantly from those seen during training. This dependency may limit its effectiveness in applications where naming conventions vary widely.
2. There are potential overfitting concerns. The authors do not provide a thorough data contamination analysis for the newly created dataset.
3. Whether the performance boosting comes from the effectiveness of the proposed pipeline or the Qwen model itself remains questionable. According to the results of ``API-Bank’’ columns in Table 5, when using the same base model, Deepseek-Coder-7B, it is not as effective as the xLAM variant. Authors could have Qwen models trained with xLAM datasets to further elaborate it.
4. The presentation of results is not well-organized. Comprehensive tables are difficult to follow and analyze, making it challenging to verify the effectiveness of each component. Authors could break down the long tables into smaller and more focused  ones. Also, for figure 3, author could have the examples shown in function masking to be larger and more compact.

**Questions:**

1. Could the authors provide a more detailed analysis of function name issues? Specifically, how many failed cases are due to function or parameter name errors? Figure 2 does not clearly convey this information. Authors may consider an error type analysis with detailed tables isolating the number of failure cases due to function name/ arguments name/ other error types.
2. It would be beneficial for the authors to analyze how many failed cases result from irrelevant function calls. The urgency of this issue is not apparent in the current version.
3. Based on the results in Table 5, it is difficult to verify the effectiveness of the proposed pipeline compared to xLAM variants. Suggestions could refer to weakness item 3.
4. While Table 1 shows that Hammer achieves impressive results on AST evaluations, the comparitive performance boosting across other benchmarks in Table 2 does not show the same level of improvement. Is there any analysis to explain this discrepancy? Authors could provide discuss potential reasons for the discrepancy and propose additional experiments or analyses that could help explain the varying levels of improvement observed.

---

> ### Author Response · Authors · 2024-11-22
> **Reply to Reviewer peJz: Part One**
>
> ### We thank Reviewer peJz for his/her constructive comments that will surely turn our paper into a better shape.
> > **Q1** The proposed methods may inherit potential biases inherent in LLMs. The sampled "Masked Data" can degrade in performance when encountering function names or parameters that differ significantly from those seen during training. This dependency may limit its effectiveness in applications where naming conventions vary widely.
>
> **A1** We sincerely apologize that our presentation might lead to misunderstanding. Indeed, the issues raised by the reviewer—"biases inherent in LLMs" and "performance degradation when naming conventions vary widely during deployment"—are critical concerns in real-world applications. However, these challenges are precisely the limitations of existing open-source models that our work aims to address or mitigate.
>
> We pointed out this issue in lines 63-65: *“Existing models tend to perform well on benchmarks that closely align with the naming conventions present in the training data but suffer notable performance declines when encountering benchmarks with differing naming styles.”* Then, we proposed masking function names and parameter names to address this, as described in lines 263-265: *“A direct approach to mitigate these issues involves minimizing the interference from function names and parameter names, while enforcing the model to comprehend the functionality and usage of candidates based on their descriptions.”* By doing so, we aim to reduce the performance drops caused by variations in naming conventions, since function descriptions provide a more flexible and less bias-prone natural language explanation.
>
> We greatly appreciate the reviewer for highlighting this concern, as it appears that we did not clearly articulate the logical connection between the motivation (lines 63-65) and the proposed method (lines 263-265). In the revision, we will ensure these sections are more coherently linked to minimize potential misunderstandings.
>
> > **Q2** There are potential overfitting concerns. The authors do not provide a thorough data contamination analysis for the newly created dataset.
>
> **A2** We sincerely appreciate the reviewer for pointing out this issue. We believe this discussion/analysis will help readers better understand the construction process of the irrelevance-augmented dataset and its impact scope on the model. We will add clarification based on the discussion below regarding the data contamination of our newly created dataset in Section 4.2.
>
> All queries/examples in the irrelevance-augmented dataset are sampled from the original xlam-function-calling-60k training set. For each sampled example, we simply remove the correct function from its candidate list and replace its label with an empty list, which indicates that all candidates are irrelevant. Therefore, the construction process of the irrelevance-augmented dataset should not introduce any additional data contamination beyond what already exists in the xlam-function-calling-60k dataset. Besides, xlam-function-calling-60k was released in January 2024, while the BFCL-V2 and Seal-Tools benchmarks used in our experiments were released in August and May 2024, respectively, both after the release of the training set. Therefore, Hammer's performance on these two benchmarks should, to some extent, address the overfitting concerns.

---

> ### Author Response · Authors · 2024-11-22
> **Reply to Reviewer peJz: Part Two**
>
> > **Q3** Whether the performance boosting comes from the effectiveness of the proposed pipeline or the Qwen model itself remains questionable. According to the results of ``API-Bank’’ columns in Table 5, when using the same base model, Deepseek-Coder-7B, it is not as effective as the xLAM variant. Authors could have Qwen models trained with xLAM datasets to further elaborate it.
>
> **A3** We sincerely appreciate the meticulous reviewer for identifying this potential issue and offering valuable suggestions, which makes our claim stronger. Following the reviewer’s advice, we first trained the Qwen models using the xLAM datasets without masking, i.e. "xxxx-xLAM" in the table below. Results with "`F1 Func-Name|F1 Func. + Args`" in each grid are presented as follows :
>
> |Models|API-Bank L-1|API-Bank L-2|Tool-Alpaca|Seal-Tools|Nexus Raven|Average|
> |:------:|:------------:|:------------:|:-----------:|:----------:|:-----------:|:-------:|
> |Qwen2-7b-xLAM  | 93.00\|83.38 | 84.01\|66.14 | 84.09\|58.18 | 95.36\|90.98 | 88.46\|72.74 | 88.98\|74.48 |
> |Hammer-7b      | 93.48\|85.79 | 82.91\|66.40 | 82.31\|59.86 | 97.44\|91.66 | 92.46\|77.35 | 89.72\|76.21 |
> |Qwen1.5-4b-xLAM| 86.68\|78.64 | 73.17\|58.32 | 84.18\|55.48 | 93.19\|88.28 | 81.28\|64.10 | 83.70\|68.96 |
> |Hammer-4b      | 91.65\|81.46 | 77.59\|61.01 | 85.09\|56.96 | 96.42\|92.45 | 81.73\|64.89 | 86.50\|71.35 |
> |Qwen2-1.5b-xLAM| 81.61\|70.93 | 80.11\|59.84 | 81.84\|53.83 | 92.77\|84.10 | 76.58\|54.90 | 82.58\|64.72 |
> |Hammer-1.5b    | 82.13\|72.30 | 79.82\|59.71 | 80.90\|53.48 | 95.59\|88.65 | 79.87\|56.88 | 83.66\|66.20 |
>
>
> And below are results on BFCL-V2:
>
> | Model | Overall Acc | AST Summary | Exec Summary | Simple AST | Multiple AST | Parallel AST | Parallel Multiple AST | Simple Exec | Multiple Exec | Parallel Exec | Parallel Multiple Exec | Irrelevance Detection | Relevance Detection |
> |:-----:|:-----:|:-----:|:-----:|:-----:|:-----:|:-----:|:-----:|:-----:|:-----:|:-----:|:-----:|:-----:|:-----:|
> | hammer-7b       | 83.92 | 78.70 | 89.71 | 69.31 | 82.52 | 78.88 | 84.08 | 91.86 | 94.00 | 88.00 | 85.00 | 72.87 | 92.68 |
> | Qwen2-7b-xLAM   | 80.42 | 72.16 | 86.50 | 66.55 | 82.05 | 67.38 | 72.67 | 83.50 | 92.00 | 88.00 | 82.50 | 76.84 | 92.68 |
> | hammer-4b       | 76.05 | 69.59 | 80.82 | 62.58 | 77.72 | 69.12 | 68.92 | 67.79 | 92.00 | 86.00 | 77.50 | 68.66 | 90.24 |
> | Qwen1.5-4b-xLAM | 74.73 | 67.75 | 80.12 | 63.88 | 77.84 | 62.62 | 66.67 | 69.00 | 90.00 | 84.00 | 77.50 | 58.20 | 97.56 |
> | Hammer-1.5b     | 73.04 | 65.52 | 75.86 | 62.34 | 72.84 | 58.75 | 68.17 | 49.93 | 92.00 | 84.00 | 77.50 | 72.18 | 92.68 |
> | Qwen2-1.5b-xLAM | 71.52 | 65.39 | 74.66 | 63.30 | 73.84 | 57.25 | 67.17 | 49.64 | 92.00 | 82.00 | 75.00 | 62.27 | 92.68 |
>
> By comparing the training results with Qwen base model in the tables above, we find that the models trained using our pipeline consistently outperform those trained on the original xLAM data across nearly all settings, which confirms the effectiveness of our pipeline.
>
> Besides, We would like to sincerely thank the reviewer for identifying this anomaly. It motivated us to further investigate the training details of xLAM-fc, which were not extensively covered in their original paper [1]. Fortunately, in their newly released paper [2] in September, they finally disclosed the data usage specifics for xLAM-fc: *“To enhance the function-calling capability of xLAM-7b-fc-r and xLAM-1b-fc-r, we employ a targeted training approach, with 50% of their training data drawn from our high-quality synthetic function-calling dataset. The remaining 50% of the training data is sampled from other tasks.”*
>
> This indicates that xLAM-fc models may still have 50% of their training data undisclosed, which we believe is likely the cause of the anomaly observed. This discovery was both surprising and exciting for us, as it suggests that our pipeline achieved superior performance across multiple benchmarks despite only utilizing half of the available data. We would like to once again express our sincere gratitude to the reviewer for their valuable input!
>
> [1] Liu, Z., Hoang, T., Zhang, J., Zhu, M., Lan, T., Kokane, S., ... & Xiong, C. (2024). Apigen: Automated pipeline for generating verifiable and diverse function-calling datasets. arXiv preprint arXiv:2406.18518.
>
> [2] Zhang, J., Lan, T., Zhu, M., Liu, Z., Hoang, T., Kokane, S., ... & Xiong, C. (2024). xlam: A family of large action models to empower ai agent systems. arXiv preprint arXiv:2409.03215.

---

> ### Author Response · Authors · 2024-11-22
> **Reply to Reviewer peJz: Part Three**
>
> > **Q4** The presentation of results is not well-organized.
>
> **A4** We greatly appreciate reviewer's valuable feedback in helping us enhancing the paper's presentation and readability. We will make the following adjustments to the existing tables and figures:
> 1. For Tables 3 and 5, we will retain only the "F1 Func. + Args" entries (the final metrics) in the main text, while moving the remaining details to the appendix.
> 2. Table 4 will be split into two separate tables (AST and Exec) to ensure the information is easier to follow and analyze.
> 3. We will refine and highlight the function masking examples in Figure 3 to improve clarity and emphasis.
>
> > **Q5** Could the authors provide a more detailed analysis of function name issues? Specifically, how many failed cases are due to function or parameter name errors?
>
> **A5** Thanks for the reviewer's suggestion. We have conducted categorical statistics for the various failed cases presented in Figure 2 to provide readers with a more detailed and comprehensive understanding. The specific results are as follows, with "`#cases with Hammer|#cases with xLAM`" in each grid:
>
> | Types | No Mask | Func. Mask | Param. Mask | All Mask |
> |--------------|:--------:|:--------:|:--------:|:--------:|
> | Correct      | 228\|187 | 213\|174 | 213\|148 | 212\|133 |
> | Func. Error  | 14\|7    | 20\|14   | 20\|13   | 23\|21   |
> | Param. Error | 49\|63   | 60\|66   | 57\|72   | 51\|70   |
> | Reject Error | 3\|37    | 1\|38    | 4\|61    | 8\|70    |
> | Total        | 294\|294 | 294\|294 | 294\|294 | 294\|294 |
>
> In this context, "Correct" refers to cases where the prediction is entirely correct; "Func. Error" indicates cases where the function itself was incorrectly selected; "Param. Error" represents cases where the function was correctly chosen, but parameter filling was erroneous; and "Reject Error" denotes cases where the model incorrectly deemed all candidates irrelevant. The table confirms xLAM's overreliance on function and parameter names, as the frequency of various errors increases significantly when function and parameter names are masked. We will replace the figure 2 with this table since it provides more information.
>
> > **Q6** It would be beneficial for the authors to analyze how many failed cases result from irrelevant function calls. The urgency of this issue is not apparent in the current version.
>
> **A6** Thank you for your thoughtful question. We analyzed the distribution of failed cases in the irrelevance detection scenario before and after using the irrelevance-augmented dataset for training the Hammer model. The results are shown in the tables below, where GT means "ground truth".
>
> Hammer-7b:
> |             | GT Irrelevant | GT Relevant |
> |---------|:----:|:----:|
> | Model Predict Irrelevant | 745 | 35 |
> | Model Predict Relevant   | 370 | 2450 |
>
> Hammer-7b-without-irrelevance-augmentation:
> |                          | GT Irrelevant | GT Relevant |
> |--------|:-----:|:----:|
> | Model Predict Irrelevant | 16 | 0 |
> | Model Predict Relevant   | 1099 | 2480 |
>
> Since xLAM has not open-sourced all its training data, the results in the table indicate that training solely on the xLAM-function-calling-60k dataset leads to poor performance in detecting irrelevant function calls (almost all ground truth irrelevant cases are predicted as relevant). We believe this strongly underscores the urgency of an open-sourced irrelevance-augmented dataset, and we will incorporate this analysis into our revision.
>
> > **Q7** While Table 2 shows that Hammer achieves impressive results on AST evaluations, the comparitive performance boosting across other benchmarks in Table 3 does not show the same level of improvement. Is there any analysis to explain this discrepancy? Authors could provide discuss potential reasons for the discrepancy and propose additional experiments or analyses that could help explain the varying levels of improvement observed.
>
> **A7** Thank you for the insightful question. Regarding the performance comparison of Hammer against different baseline models across various benchmarks, we acknowledge that the level of improvement can indeed exhibit variation. Our insights into the potential reasons are as follows:
> 1. As demonstrated in Table 1 of our paper, the performance of baseline models inherently fluctuates across different benchmarks. This instability is exactly what our work is trying to solve.
> 2. Certain benchmarks, such as Tool-Alpaca, contain function candidates in the test set with very concise descriptions. In such cases, the information Hammer can extract from the descriptions is inherently limited (often overlapping significantly with the function names). Consequently, the performance improvement achieved by Hammer on Tool-Alpaca is smaller compared to other benchmarks. This observation underscores the importance of providing clear and comprehensive descriptions for each function in the library when designing function-calling tasks.

---

> ### Comment · Reviewer_peJz · 2024-11-30
>
> Thank you for your feedback. Most of the concerns are solved. I have raised my score from 5 to 6.

---

### Author Response · Authors · 2024-11-27
**Meta Responses**

# Meta Responses
### We are delighted to receive positive feedback from all the reviewers, and thank the reviewers for their valuable suggestions. Below, we summarized the modifications in our current revision that are marked with cyan.

1. Strengthen the coherence between existing issues and our intention for this research in Section 1 (lines 62-65).
2. Add clarification regarding the data contamination of our newly created dataset in Secton 4.2 (lines 322-327) and Appendix C.
3. Add performance of Qwen models trained with xLAM datasets in Table 6 to further verify the effectiveness of the proposed pipeline.
4. We retain only the "F1 Func. + Args" entries (the final metrics) in the main text (see Table 3 and 6 in revision), while moving the remaining details to the appendix E.
5. Separate the AST and Exec evaluation (See Table 4 and 5 in revision) to ensure the information is easier to follow and analyze.
6. In Appendix B, we provide categorical statistics for the "various failed cases presented in Figure 2".
7. In Appendix G, we provide the analysis and distribution of failed cases in the irrelevance detection scenario before and after using the irrelevance-augmented dataset for training the Hammer model.
8. Add clarification about our masking ratio chosen in Section 4.1 (lines 287-290).
9. In Appendix A, we provide categorical statistics for the various types of failed cases appeared in the Seal-Tools and Nexus Raven benchmarks where xLAM performs poorly, to provide readers with a more detailed and comprehensive understanding.
10. Add on-device experiments where hammer-7b is deployed in a mobole phone, and analysis it performance in Section 5 (lines 376-377) and Appendix H.
11. Update Section 5.1 (lines 351-354) to include a discussion on ToolBench, providing further clarification on the rationale behind our benchmark selection.
12. Add discussion of potential failure modes or edge cases where function masking might perform worse than traditional approaches in Section 5 (lines 333-334) and Appendix D.
13. Add details of the random string generation process for masking in Section 4.1 (lines 268-269).

---

### Meta-Review · Area_Chair_chpJ · 2024-12-21

**Metareview:**

This paper introduces a new family of models, Hammer, which focuses on enhancing the robustness of function-calling capabilities in large language models through a novel masking approach. The authors identify a critical issue with existing models that overfit specific naming conventions, resulting in inconsistent performance. Hammer mitigates this by employing function masking and augmenting datasets to better handle irrelevance detection, achieving state-of-the-art results across multiple benchmarks. Key contributions include the open-sourcing of datasets and tuning frameworks, which have practical implications for on-device deployment.

Strengths of the paper include its focus on a significant real-world problem, robust evaluations across benchmarks, and innovative techniques like function masking. The paper also effectively addresses concerns about performance generalization and includes comprehensive ablation studies. Weaknesses lie in limited exploration of certain edge cases, such as scenarios with sparse function descriptions or broader applications of the technique. While the discussion around hardware efficiency is present, further clarification about resource constraints could enhance practical applicability.

The decision to accept this paper is based on its innovative methodology, significant practical implications, and thorough experimental validation. The results provide a substantial advancement in the robustness of function-calling models, particularly for on-device applications, making this work a valuable contribution to the community.

**Additional Comments On Reviewer Discussion:**

The authors demonstrated a high level of responsiveness during the rebuttal period, addressing all major concerns raised by reviewers. Discussions on hardware efficiency and potential limitations of the masking technique were particularly constructive. These updates, including additional experiments and clarifications, solidified the paper's claims and addressed reviewer feedback effectively, further justifying the recommendation for acceptance.

---

### Decision · Program_Chairs · 2025-01-22

Accept (Spotlight)